# Zero-Sacrifice Persistent-Robustness Adversarial Defense for Pre-Trained Encoders

**Zhuxin Lei**[a,b,c]**, Ziyuan Yang**[a,b,c,*]**, Yi Zhang**[a,b,c,*]
[a] School of Cyber Science and Engineering, Sichuan University
[b] Key Laboratory of Data Protection and Intelligent Management, Ministry of Education, Sichuan University
[c] Tianfu Jiangxi Laboratory
`2022141530064@stu.scu.edu.cn, cziyuanyang@gmail.com,`
`yzhang@scu.edu.cn`

## Abstract

The widespread use of publicly available pre-trained encoders from self-supervised learning (SSL) has exposed a critical vulnerability: their susceptibility to downstream-agnostic adversarial examples (DAEs), which are crafted without knowledge of the downstream tasks but capable of misleading downstream models. While several defense methods have been explored recently, they rely primarily on task-specific adversarial fine-tuning, which inevitably limits generalizability and causes catastrophic forgetting and deteriorates benign performance. Different with previous works, we propose a more rigorous defense goal that requires only a single tuning for diverse downstream tasks to defend against DAEs and preserve benign performance. To achieve this defense goal, we introduce *Ze*ro-Sacrifice *P*ersistent-Robustness *A*dversarial *D*efense (*ZePAD*), which is inspired by the inherent sensitivity of neural networks to data characteristics. Specifically, ZePAD is a dual-branch structure, which consists of a Multi-Pattern Adversarial Enhancement Branch (MPAE-Branch) that uses two adversarially fine-tuned encoders to strengthen adversarial resistance. The Benign Memory Preservation Branch (BMP-Branch) is trained on local data to ensure adversarial robustness does not compromise benign performance. Surprisingly, we find that ZePAD can directly detect DAEs by evaluating branch confidence, without introducing any adversarial exsample identification task in training. Notably, by enriching feature diversity, our method enables a single adversarial fine-tuning to defend against DAEs across downstream tasks, thereby achieving persistent robustness. Extensive experiments on 11 SSL methods and 6 datasets validate its effectiveness. In certain cases, it achieves a 29.20% improvement in benign performance and a 73.86% gain in adversarial robustness, highlighting its zero-sacrifice property. To facilitate reproducibility, the code is publicly available[1].

## 1 Introduction

Training robust models is resource-intensive. Consequently, using public pre-trained models from self-supervised learning (SSL), such as CLIP (Radford et al., 2021) and GPT (Brown, 2020), is a common paradigm for downstream tasks. These encoders learn powerful representations, fostering advancements in fields like classification (Huang et al., 2023) and segmentation (Liu et al., 2023).

However, these encoders are vulnerable to Downstream-Agnostic Adversarial Examples (DAEs) (Liu et al., 2022), which are crafted without knowledge of the downstream tasks but capable of misleading downstream models. Even when the model has been task-specific tuned on local data, the attack remains effective. For example, PAP (Ban & Dong, 2022) and AdvEncoder (Zhou et al., 2023) can generate effective DAEs without knowledge of the pre-training or downstream data.

---

[*]Ziyuan Yang and Yi Zhang are the corresponding authors.
[1]https://github.com/Lawliet0o/ZePAD

To address this security concern, researchers have proposed several secure fine-tuning methods against DAEs, but these methods lead to a significant drop in benign performance, making them impractical. For instance, Gen-AF (Zhou et al., 2024) improves robust accuracy from 35.58% to 62.28% but reduces benign accuracy from 81.99% to 68.69%. Meanwhile, these methods require re-tuning for each task. This leads us to ask:

*"Is it possible to achieve a zero-sacrifice persistent-robustness defense that requires only a single tuning for various downstream tasks, without compromising the performance of benign samples?"*

We find that neural networks inherently exhibit higher confidence in inputs that resemble the training data, a behavior attributed to the memorization of the data's characteristics Zhang et al. (2016). Inspired by this tendency, we shift our focus from traditional adversarial fine-tuning to evaluating the confidence levels of different encoders.

This insight motivates us to rethink the defense process, shifting the focus from finding a tradeoff between adversarial robustness and performance decline to achieving a zero-sacrifice approach. We propose *Ze*ro-Sacrifice *L*ifelong *A*dversarial *D*efense (***ZePAD***), which uses two branches to enhance robustness while maintaining or even improving benign performance: ***i) Multi-Pattern Adversarial Enhancement Branch (MPAE-Branch):*** To counter DAEs that exploit specific patterns, this branch adversarially fine-tunes encoders trained with diverse SSL methods. This diversity makes it difficult for attackers to find common vulnerabilities. ***ii) Benign Memory Preservation Branch (BMP-Branch):*** To mitigate the performance drop on clean data from adversarial fine-tuning, this branch is trained only on benign samples, preserving the model's sensitivity to them.

To effectively integrate the different branches and achieve zero-sacrifice, we propose a ***R***obust ***F***ederal ***D***ecision ***M***echanism (***RFDM***). Each branch's confidence adapts to the alignment between the input and its training dataset distribution, and RFDM makes a decision based on this evaluation. Specifically, for benign samples, the BMP-Branch assigns higher confidence than the other branches, as the latter are adversarially fine-tuned and more biased towards adversarial exsamples. Since adversarial exsamples were not encountered during BMP-Branch training, they fall outside its distribution, resulting in lower confidence. Similarly, the MPAE-Branch, trained with adversarial examples, exhibits higher confidence when handling such samples.

Existing defense methods typically require task-specific adversarial fine-tuning. In this paper, we show that introducing diversification through multiple heterogeneous encoders enhances robustness beyond single-encoder fine-tuning. As a result, ZePAD requires only a single fine-tuning process to provide strong, versatile representations across all downstream tasks, making it a persistent-robustness defense method. Surprisingly, the aforementioned inherent characteristics of neural networks allow ZePAD to detect adversarial exsamples based solely on confidence scores, without requiring specific training for this purpose. Our main contributions are:

- We propose a zero-sacrifice, persistent-robustness adversarial defense method that not only maintains but also improves benign performance, while enhancing adversarial robustness. Besides, our method only requires a single-time adversarial tuning to achieve robust performance across various downstream tasks.

- Although adversarial exsample detection is not the focus of our work, our method can directly detect adversarial exsamples, without requiring any specific training for this purpose.

- Extensive experiments across 11 SSL methods and 6 datasets validate ZePAD's effectiveness, showing improvements as high as 29.20% in benign performance and 73.05% in robust performance in some scenarios.

## 2 RELATED WORKS

***Self-Supervised Learning.*** SSL pre-trains a generic encoder on vast unlabeled data for various downstream tasks. This paradigm overcomes labeled data limitations, allowing users to adapt powerful pre-trained encoders through fine-tuning without training from scratch. Common adaptation methods include linear evaluation (Aghajanyan et al., 2020; Kumar et al., 2022), which adjusts only the last layer, and full fine-tuning (Hendrycks et al., 2019; Miller et al., 2021), which tunes all layers.

***Adversarial Attacks.*** Deep neural networks (DNNs) are vulnerable to adversarial examples (Goodfellow et al., 2014; Hu et al., 2022), including universal adversarial perturbations (UAPs) Co et al. (2019) where a single perturbation affects multiple samples. Attacks are categorized as white-box (Naseer et al., 2020), where the attacker has full model access, or black-box (Lin et al., 2024), with access only to inputs and outputs. Recent research (Ban & Dong, 2022; Zhou et al., 2023; Zhang et al., 2022) explores attacks on pre-trained encoders, which are downstream-agnostic and thus pose a significant security risk to the pre-trained paradigm.

***Adversarial Defenses.*** Adversarial defense methods include input filtering (Guo et al., 2017) and enhancing model robustness, often through adversarial training (Zhu & Gupta, 2017; Zhou et al., 2024; Li et al., 2024; Liu et al., 2021). However, it typically harms generalization, reducing performance on clean samples. While recent work (Lamb et al., 2019; Zhou et al., 2024) aims to balance robustness and generalization, no existing method enhances robustness without compromising—or while even improving—generalization ability. The detailed related works are discussed in ***Appendix A***.

## 3 PROBLEM FORMULATION AND THREAT MODELING

### 3.1 PROBLEM STATEMENT

Let $\mathcal{E}_\theta$ be a pre-trained encoder, parameterized by $\theta$. We define $\mathcal{D}_p$ as the dataset used to pre-train $\mathcal{E}_\theta$, where the benign input is represented as $(x, y) \in \mathcal{D}_p$, with $x$ as the input and $y$ as the corresponding label. The output feature of $\mathcal{E}_\theta$ is denoted as $v = \mathcal{E}_\theta(x)$, where $v \in V$ and $V$ represents the feature space. The downstream model is denoted as $\mathcal{F}_\phi$, where $\phi$ represents its parameters. The input to $\mathcal{F}_\phi$ is the output of $\mathcal{E}_\theta$, and the process can be formulated as $\mathcal{F}_\phi(v)$.

In adversarial attacks, an attacker aims to exploit vulnerabilities in $\mathcal{E}_\theta$ by adding small perturbations $\delta$ to the input $x$. The perturbation $\delta$ is constrained by an upper bound $\epsilon$ based on the $l_p$-norm, which ensures the noise remains imperceptible. The adversarial attack can be described as:

$$\mathcal{E}_\theta(x) \neq \mathcal{E}_\theta(x + \delta), \text{ subject to } \|\delta\|_p \leq \epsilon. \tag{1}$$

The ultimate goal of the attacker is to deceive the downstream model $\mathcal{F}_\phi$. Specifically, for a classification task, the attacker aims to manipulate $\mathcal{F}_\phi$ to output different predictions for benign and adversarial examples. This can be formalized as follows:

$$\mathcal{F}_\phi(\mathcal{E}_\theta(x)) \neq \mathcal{F}_\phi(\mathcal{E}_\theta(x + \delta)), \text{ subject to } \|\delta\|_p \leq \epsilon. \tag{2}$$

As defenders, our goal is to create a defense mechanism that prevents the adversarial attack from succeeding while preserving or improving benign performance. Specifically, we aim to ensure that the predictions for both the original input $x$ and the adversarial exsample $x + \delta$ remain the same.

### 3.2 THREAT MODEL

This paper investigates the vulnerability of pre-trained encoders in downstream classification tasks. We assume attackers exploit publicly available encoders, whcih can be obtained from the public platforms, to craft adversarial examples that transfer to any downstream model utilizing the encoder. These attacks are non-targeted, universal across downstream tasks, effective even after fine-tuning, and stealthy, requiring perturbations to remain visually imperceptible.

For defenders, they cannot control the pre-training stage and are unaware of the original training data or SSL method used in pre-training. However, they have full access to the released encoder and complete control over downstream fine-tuning, including architectural adaptation.

We propose a stronger defender's objective: improving downstream robustness without sacrificing benign performance. Our defender's goals can be summarized as: ***1) Zero-Sacrifice Accuracy:*** Maintain (or even improve) benign performance without degrading clean accuracy; ***2) Adversarial Robustness:*** Resist adversarial examples generated from pre-trained encoders and maintain correct predictions; ***3) Resource Efficiency:*** Avoid reliance on external data or additional computational costs beyond standard fine-tuning; ***4) Persistent Robustness:*** Enable a single adversarially-tuned encoder to remain robust across diverse downstream tasks without task-specific retraining.

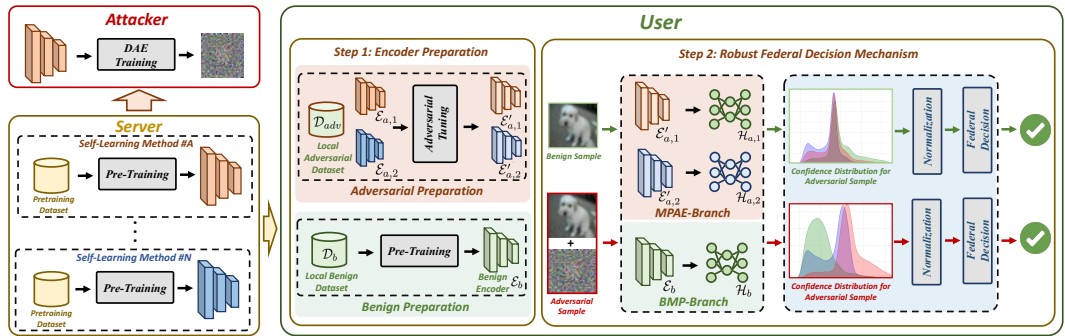

Figure 1: The overview of the proposed method. The red and green arrows denote the attack flow and normal flow in the inference stage, respectively.

## 3.3 CHALLENGES

Unlike previous works, we aim to enhance adversarial robustness without sacrificing benign performance, and even expect to improve it. Furthermore, we aim to establish a persistent-robustness paradigm that enables a one-time tuning process while ensuring robustness across all downstream tasks. However, just re-tuning the pre-trained encoder may compromise its powerful feature extraction capabilities. Hence, achieveing our goal is challenging, which can be outlined as:

(1) *Challenge I: Zero-Sacrifice Conflict:* Existing adversarial fine-tuning methods inevitably degrade benign accuracy due to the inherent conflict between robustness optimization and clean performance. Our work targets the more challenging zero-sacrifice objective that achieving robustness improvement without harming benign accuracy, despite limited access to the pre-training process.

(2) *Challenge II: Persistent Robustness Across Tasks:* Most defenses require task-specific tuning and cannot generalize across downstream tasks. We introduce persistent robustness, aiming for "tune once, defend all" capability, enabling a single tuned encoder to remain robust across diverse downstream tasks without re-training or task-specific assumptions.

The detailed threat model and challenges are provided in *Appendix B*.

## 4 METHODOLOGY

### 4.1 ZEPAD: A COMPLETE ILLUSTRATION

In this section, we introduce ZePAD, the first zero-sacrifice, persistent-robustness defense method specifically designed for pre-trained encoders. The pipeline of the proposed method is illustrated in Figure 1, and it comprises two primary steps: encoder preparation and RFDM. These steps are elaborated upon in the following subsections.

### 4.1.1 STEP 1: ENCODER PREPARATION

The core motivation behind ZePAD in achieving persistent-robustness zero-sacrifice defense against DAEs is to leverage the inherent confidence bias within neural networks. Therefore, the first step in ZePAD is encoder preparation. Since the defender can modify the local encoder, the goal of this step is to build a feature extractor with both strong representation capabilities and robust adversarial resilience. Specifically, our encoder consists of the MPAE-Branch and the BMP-Branch, which are designed to enhance adversarial robustness and improve benign performance, respectively.

**MPAE-Branch:** DAEs exploit the model's sensitivity to specific patterns and textures, but it is hard to learn common sensitive patterns across multiple encoders, trained with different SSL methods. The diversity in feature representations learned by different encoders makes it difficult for DAEs to pinpoint shared vulnerabilities. This significantly complicates the task of exploiting common weaknesses across all encoders. Hence, our MPAE-Branch consists of two public pre-trained encoders.

To ensure adversarial robustness, both encoders must be fine-tuned to strengthen their resilience to adversarial attacks. However, pre-trained encoders are highly sensitive to fine-tuning, which presents a dilemma: how to improve adversarial robustness while preserving the original pre-trained knowledge. To address this challenge, we propose a hybrid loss that strikes a balance between preserving the pre-trained encoders' knowledge and enhancing the adversarial robustness. The loss function is formulated as follows:

$$\mathcal{L} = \mathcal{L}_c + \lambda \mathcal{L}_f, \tag{3}$$

where $\mathcal{L}_c$ represents the classification loss of adversarial exsamples, and $\mathcal{L}_f$ represents the encoding feature loss of adversarial exsamples. $\lambda$ is a hyperparameter. Then, $\mathcal{L}_c$ is calculated as follows:

$$\mathcal{L}_c(x, y) = \mathcal{L}_{CE}\left(\mathcal{F}_\phi\left(\mathcal{E}_\theta\left(x + \delta\right)\right), y\right), \tag{4}$$

where $\mathcal{E}$ denotes the encoder, and $\mathcal{F}$ represents the downstream classification model. $x$ and $y$ stand for the data and labels, respectively. To better describe the loss caused by encoding features, we first define the distance between two encoding features. We use cosine distance to measure the feature differences between them. Additionally, we remove the nearest features to prevent local high-density data from overly influencing the loss value and to better describe the global structure. This process is formulated as follows:

$$D_{ij} = \frac{2 - \left(\cos(x_i, x_j) - \rho_{x_j}\right)}{\sum_{k=1, k \neq j}^{N}\left(2 - \left(\cos(x_i, x_k)\right) - \rho_{x_k}\right)}, \tag{5}$$

where $\cos(x_i, x_j)$ denotes the cosine distance between $x_i$ and $x_j$, and $\rho_{x_j}$ denotes the minimum distance between $x_j$ and all samples in the training dataset. Then, the loss can be formulated as:

$$\mathcal{L}_f = \sum_i \sum_j \left[ D_{ij}^b \log\left(\frac{D_{ij}^b}{D_{ij}^a}\right) + \left(1 - D_{ij}^b\right) \log\left(\frac{1 - D_{ij}^b}{1 - D_{ij}^a}\right) \right], \tag{6}$$

where $D_{ij}^a$ and $D_{ij}^b$ represent the distances between the adversarial exsample pair $(x_i, x_j)$ and their corresponding benign sample pair, respectively. This makes the feature distribution of adversarial exsamples after encoding similar to that of benign samples.

**BMP-Branch:** To address this challenge, we introduce a benign branch. In this step, we train an additional encoder locally using only benign data. By focusing on local data, the benign branch can memorize and capture crucial features that are essential for maintaining strong performance on clean, non-adversarial inputs.

Overall, our encoder $\mathcal{E}$ consists of three sub-encoders: $\mathcal{E}_{a,1}$, $\mathcal{E}_{a,2}$, and $\mathcal{E}_b$. Here, $\mathcal{E}_{a,1}$ and $\mathcal{E}_{a,2}$ represent the publicly shared and adversarially tuned pre-trained encoders, while $\mathcal{E}_b$ refers to the locally trained benign encoder. As assumed, the downstream task is a classification problem. The classification heads can be trained based on these sub-encoders using the local data. This process allows us to obtain classification heads $\mathcal{H}_{a,1}$, $\mathcal{H}_{a,2}$, and $\mathcal{H}_b$, which are responsible for mapping the features from $\mathcal{E}_{a,1}$, $\mathcal{E}_{a,2}$, and $\mathcal{E}_b$, respectively. A key challenge now is how to effectively combine these sub-encoders to achieve the zero-sacrifice goal.

### 4.1.2 STEP 2: ROBUST FEDERAL DECISION MECHANISM

Our method builds on the observation that neural networks implicitly estimate posterior confidence. Consequently, they assign higher confidence to samples aligned with their training distribution and lower confidence to out-of-distribution or adversarial inputs. This inherent behavior leads to natural confidence separation between branches for different kinds of data. To support our observation, the theoretical proof can be found in ***Appendix C***.

To fully leverage this observation, we utilize two branches trained for different sample types: the MPAE-Branch for adversarial exsamples and the BMP-Branch for benign ones. This design leverages the principle that neural networks show higher confidence on inputs consistent with their training data (Carlini et al., 2019). The MPAE-Branch is adversarially fine-tuned for robustness, while the BMP-Branch, trained solely on benign samples, is highly sensitive to clean data.

Our experiments also support our assumption, as illustrated in Figure 2. On benign samples, the BMP-Branch consistently exhibits higher confidence. Conversely, the MPAE-Branch's adversarial training, while enhancing robustness, slightly compromises its generalization on clean data, resulting in lower confidence.

Table 1: BA (%) comparison between the baseline method and ZePAD in the semi-black-box setting, where $\mathcal{D}_p$, $\mathcal{D}_f$ and $\mathcal{D}_d$ denote the pre-training, adversarialt adversarial training and downstream datasets, respectively.

| $\mathcal{D}_p$ | $\mathcal{D}_f$ and $\mathcal{D}_d$ | Method | W-MSE | BYOL | NNCLR | SimCLR | MoCo2+ | MoCo3 | SupCon | RESSL | DINO | SwAV | VibCreg | AVG |
|---|---|---|---|---|---|---|---|---|---|---|---|---|---|---|
| CIFAR10 | CIFAR10 | baseline | 90.17 | 94.39 | 93.77 | 93.56 | 95.03 | 94.71 | 96.22 | 92.73 | 91.43 | 92.29 | 92.39 | 93.34 |
| | | ours | 93.42 | 93.49 | 93.96 | 95.05 | 93.79 | 94.27 | 92.39 | 92.87 | 93.36 | 93.51 | 93.12 | **93.57** |
| | | Δ | +3.25 | -0.90 | +0.19 | +1.49 | -1.24 | -0.44 | -3.83 | +0.14 | +1.93 | +1.22 | +0.73 | +0.23 |
| | STL10 | baseline | 78.49 | 83.08 | 83.09 | 81.94 | 84.46 | 84.00 | 85.29 | 81.87 | 83.54 | 81.53 | 81.64 | 82.63 |
| | | ours | 81.43 | 82.30 | 82.52 | 84.38 | 81.52 | 83.51 | 82.90 | 82.10 | 81.42 | 81.87 | 83.65 | **82.51** |
| | | Δ | +2.94 | -0.78 | -0.57 | +2.44 | -2.94 | -0.49 | -2.39 | +0.23 | -2.12 | +0.34 | +2.01 | -0.12 |
| | ANIMALS10 | baseline | 80.39 | 90.22 | 91.22 | 90.82 | 88.03 | 88.05 | 88.23 | 87.86 | 85.01 | 89.16 | 94.70 | 88.52 |
| | | ours | 97.35 | 98.29 | 97.84 | 97.92 | 98.56 | 98.41 | 97.79 | 98.67 | 97.76 | 98.07 | 98.86 | **98.14** |
| | | Δ | +16.96 | +8.07 | +6.62 | +7.10 | +10.53 | +10.36 | +9.56 | +10.81 | +12.75 | +8.91 | +4.16 | **+9.62** |
| | GTSRB | baseline | 73.76 | 75.99 | 79.06 | 78.61 | 76.37 | 81.11 | 81.65 | 76.03 | 80.63 | 85.51 | 82.25 | 79.18 |
| | | ours | 95.34 | 93.58 | 94.20 | 94.99 | 94.78 | 94.38 | 94.23 | 94.12 | 94.41 | 95.54 | 94.09 | **94.51** |
| | | Δ | +21.58 | +17.59 | +15.14 | +16.38 | +18.41 | +13.27 | +12.58 | +18.09 | +13.78 | +10.03 | +11.84 | **+15.34** |
| CIFAR10 | ImageNet20 | baseline | 59.01 | 67.47 | 65.94 | 57.74 | 66.22 | 65.00 | 58.87 | 61.95 | 66.45 | 63.01 | 62.31 | 63.09 |
| | | ours | 73.93 | 72.88 | 73.09 | 74.08 | 74.45 | 74.45 | 72.54 | 72.39 | 73.35 | 72.94 | 73.41 | **73.41** |
| | | Δ | +14.92 | +5.41 | +7.15 | +16.34 | +8.23 | +9.45 | +13.67 | +10.44 | +6.90 | +9.93 | +11.10 | **+10.32** |
| | SVHN | baseline | 57.92 | 58.59 | 66.43 | 64.81 | 64.67 | 64.32 | 69.88 | 67.65 | 70.38 | 72.57 | 73.26 | 66.41 |
| | | ours | 94.08 | 94.70 | 94.84 | 94.92 | 95.39 | 94.75 | 94.22 | 94.80 | 94.40 | 94.52 | 93.91 | **94.59** |
| | | Δ | +36.16 | +36.11 | +28.41 | +30.11 | +30.72 | +30.43 | +24.34 | +27.15 | +24.02 | +21.95 | +20.65 | **+28.19** |
| ImageNet | CIFAR10 | baseline | 56.52 | 68.60 | 72.38 | 70.56 | 70.35 | 72.37 | 71.35 | 71.20 | 71.75 | 68.46 | 73.35 | 69.72 |
| | | ours | 93.03 | 93.05 | 93.39 | 93.07 | 93.46 | 93.88 | 93.44 | 92.43 | 93.05 | 93.02 | 93.12 | **93.18** |
| | | Δ | +36.51 | +24.45 | +21.01 | +22.51 | +23.11 | +21.51 | +22.09 | +21.23 | +21.30 | +24.56 | +19.77 | **+23.46** |
| | STL10 | baseline | 51.99 | 63.37 | 65.41 | 66.14 | 64.78 | 63.21 | 64.79 | 64.89 | 64.37 | 62.90 | 68.76 | 63.69 |
| | | ours | 80.08 | 76.82 | 81.14 | 82.07 | 81.60 | 82.94 | 79.35 | 80.22 | 80.77 | 80.58 | 81.35 | **80.63** |
| | | Δ | +28.09 | +13.45 | +15.73 | +15.93 | +16.82 | +19.73 | +14.56 | +15.33 | +16.40 | +17.68 | +12.59 | **+16.94** |
| | ANIMALS10 | baseline | 48.53 | 63.82 | 68.81 | 70.69 | 65.86 | 68.31 | 74.65 | 64.75 | 62.19 | 61.75 | 76.57 | 65.99 |
| | | ours | 97.28 | 97.70 | 98.30 | 97.86 | 97.70 | 98.35 | 98.55 | 98.05 | 98.18 | 97.73 | 98.61 | **98.03** |
| | | Δ | +48.75 | +33.88 | +29.49 | +27.17 | +31.84 | +30.04 | +23.90 | +33.30 | +35.99 | +35.98 | +22.04 | **+32.03** |
| | GTSRB | baseline | 52.17 | 67.65 | 72.63 | 69.36 | 67.21 | 74.88 | 78.05 | 70.09 | 68.94 | 72.17 | 77.96 | 70.10 |
| | | ours | 94.56 | 92.84 | 94.34 | 94.98 | 94.12 | 94.33 | 94.87 | 93.79 | 94.55 | 94.76 | 94.44 | **94.33** |
| | | Δ | +42.39 | +25.19 | +21.71 | +25.62 | +26.91 | +19.45 | +16.82 | +23.70 | +25.61 | +22.59 | +16.48 | **+24.22** |
| | ImageNet20 | baseline | 35.64 | 48.76 | 56.18 | 56.44 | 52.35 | 54.36 | 47.40 | 51.49 | 51.13 | 47.84 | 54.06 | 50.51 |
| | | ours | 72.80 | 72.75 | 72.99 | 73.97 | 75.34 | 75.69 | 71.20 | 73.16 | 72.33 | 71.91 | 69.81 | **72.90** |
| | | Δ | +37.16 | +23.99 | +16.81 | +17.53 | +22.99 | +21.33 | +23.80 | +21.67 | +21.20 | +24.07 | +15.75 | **+22.39** |
| | SVHN | baseline | 50.45 | 62.86 | 68.40 | 69.25 | 62.50 | 64.31 | 67.40 | 63.95 | 71.31 | 66.72 | 69.51 | 65.15 |
| | | ours | 94.16 | 93.60 | 94.05 | 94.92 | 95.20 | 94.75 | 94.36 | 94.57 | 94.21 | 94.03 | 94.03 | **94.35** |
| | | Δ | +43.71 | +30.74 | +25.65 | +25.67 | +32.70 | +30.44 | +26.96 | +30.62 | +22.90 | +27.31 | +24.52 | **+29.20** |

On adversarial exsamples, the roles reverse. The BMP-Branch's confidence drops significantly as it is vulnerable to the versatile attacks. In contrast, the MPAE-Branch maintains the highest confidence, as its adversarial fine-tuning makes it resilient to these specific inputs.

This observed phenomenon, where the branches exhibit complementary strengths, is the foundation of our proposed approach. We leverage this insight to combine their outputs, aiming to maximize both adversarial robustness and performance on benign data. The decision process is formulated as follows:

$$\mathcal{W}_i = \frac{c_i}{\sum_{k=1}^{3} c_k}, \qquad (7)$$

where $\mathcal{W}_i$ represents the confidence score of the $i$-th branch, while $c_i$ represents the confidence value of the $i$-th branch. Specifically, the calculation method for $c_i$ is as follows:

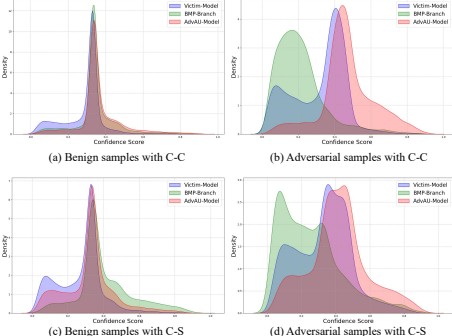

(a) Benign samples with C-C  (b) Adversarial samples with C-C

(c) Benign samples with C-S  (d) Adversarial samples with C-S

Figure 2: The confidence distributions of each branch under different conditions. C-C indicates fine-tuning on CIFAR10 and downstream testing on CIFAR10; C-S indicates fine-tuning on CIFAR10 and downstream testing on STL10.

$$c_i = m_i e^{3(m_i - 0.5)}, \qquad (8)$$

where $m_i$ represents the maximum value in the probability vector output of $i$-th branch. However, we argue that the confidence value does not change linearly with the maximum probability. Instead, we posit the maximum probability value is positively correlated with the confidence value. Besides, as the maximum probability value approaches 1, the rate at which the confidence value increases should accelerate. To help readers follow our work, the algorithm can be found in ***Appendix D***.

## 5 EXPERIMENTS

### 5.1 EXPERIMENTAL SETTING

**Attacks.** We evaluate the effectiveness of our work using the following five general adversarial attacks: AdvEncoder (Zhou et al., 2023), PAP (Ban & Dong, 2022), UAP (Moosavi-Dezfooli et al., 2017), UAPGD (Deng & Karam, 2020) and SSP (Naseer et al., 2020).

Table 2: RA (%) comparison between the baseline method and ZePAD in the semi-black-box setting, where $D_p$, $D_f$ and $D_d$ denote thepre-training, adversarialt adversarial training and downstream datasets, respectively

| $D_p$ | $D_f$ and $D_d$ | Method | W-MSE | BYOL | NNCLR | SimCLR | MoCo2+ | MoCo3 | SupCon | RESSL | DINO | SwAV | VibCreg | AVG |
|---|---|---|---|---|---|---|---|---|---|---|---|---|---|---|
| CIFAR10 | CIFAR10 | baseline | 10.09 | 12.47 | 10.71 | 47.13 | 45.01 | 36.86 | 16.00 | 12.05 | 10.46 | 31.67 | 14.07 | 22.41 |
| | | ours | 86.84 | 80.18 | 79.84 | 74.70 | 69.70 | 72.49 | 77.99 | 84.58 | 80.44 | 74.61 | 79.62 | **78.27** |
| | | Δ | +76.75 | +67.71 | +69.13 | +27.57 | +24.69 | +35.63 | +61.99 | +72.53 | +69.98 | +42.94 | +65.55 | **+55.86** |
| | STL10 | baseline | 41.16 | 46.85 | 44.05 | 67.52 | 65.55 | 64.28 | 56.89 | 51.69 | 31.77 | 59.55 | 43.07 | 52.03 |
| | | ours | 70.28 | 71.30 | 72.66 | 77.34 | 74.01 | 74.89 | 75.17 | 73.74 | 71.73 | 72.12 | 71.18 | **73.13** |
| | | Δ | +29.12 | +24.45 | +28.61 | +9.82 | +8.46 | +10.61 | +18.28 | +22.05 | +39.96 | +12.57 | +28.11 | **+21.09** |
| | ANIMALS10 | baseline | 48.31 | 49.25 | 49.23 | 63.48 | 63.03 | 60.32 | 58.48 | 54.92 | 35.46 | 63.75 | 50.19 | 54.22 |
| | | ours | 91.94 | 90.49 | 86.71 | 87.94 | 88.34 | 84.58 | 88.52 | 89.02 | 89.01 | 87.88 | 88.65 | **88.46** |
| | | Δ | +43.63 | +41.24 | +37.48 | +24.46 | +25.31 | +24.26 | +30.04 | +34.10 | +53.55 | +24.13 | +38.46 | **+34.24** |
| | GTSRB | baseline | 10.85 | 12.41 | 11.29 | 24.63 | 27.02 | 27.60 | 13.22 | 12.37 | 10.62 | 27.82 | 11.56 | 17.22 |
| | | ours | 75.00 | 79.23 | 73.95 | 74.59 | 74.15 | 73.54 | 72.15 | 72.47 | 80.89 | 76.04 | 72.18 | **74.93** |
| | | Δ | +64.15 | +66.82 | +62.66 | +49.96 | +47.13 | +45.94 | +58.93 | +60.10 | +70.27 | +48.22 | +60.62 | **+57.71** |
| | ImageNet20 | baseline | 27.60 | 34.41 | 29.82 | 38.24 | 42.87 | 41.60 | 28.97 | 35.10 | 21.24 | 37.69 | 27.13 | 33.15 |
| | | ours | 60.07 | 64.84 | 64.54 | 61.69 | 62.60 | 62.68 | 63.74 | 63.55 | 65.17 | 62.46 | 55.99 | **62.48** |
| | | Δ | +32.47 | +30.43 | +34.72 | +23.45 | +19.73 | +21.08 | +34.77 | +28.45 | +43.93 | +24.77 | +28.86 | **+29.33** |
| | SVHN | baseline | 9.16 | 11.68 | 14.99 | 14.58 | 21.51 | 16.38 | 11.10 | 9.27 | 8.95 | 16.08 | 10.55 | 13.11 |
| | | ours | 75.65 | 89.17 | 88.52 | 91.26 | 86.68 | 90.75 | 90.53 | 86.67 | 91.39 | 86.67 | 79.43 | **86.97** |
| | | Δ | +66.49 | +77.49 | +73.53 | +76.68 | +65.17 | +74.37 | +79.43 | +77.40 | +82.44 | +70.59 | +68.88 | **+73.86** |
| ImageNet | CIFAR10 | baseline | 11.78 | 11.78 | 12.80 | 24.31 | 16.49 | 12.66 | 14.00 | 11.33 | 15.80 | 12.44 | 20.42 | 14.89 |
| | | ours | 74.00 | 66.69 | 66.03 | 78.07 | 68.01 | 69.37 | 71.21 | 81.77 | 78.72 | 86.01 | 67.22 | **73.37** |
| | | Δ | +62.22 | +54.91 | +53.23 | +53.76 | +51.52 | +56.71 | +57.21 | +70.44 | +62.92 | +73.57 | +46.80 | **+58.48** |
| | STL10 | baseline | 26.76 | 33.02 | 22.42 | 46.23 | 40.43 | 42.37 | 25.12 | 41.98 | 33.68 | 42.77 | 40.84 | 35.97 |
| | | ours | 70.35 | 67.80 | 64.67 | 73.26 | 74.30 | 76.78 | 67.00 | 71.90 | 70.18 | 75.18 | 69.36 | **70.98** |
| | | Δ | +43.59 | +34.78 | +42.25 | +27.03 | +33.87 | +34.41 | +41.88 | +29.92 | +36.50 | +32.41 | +28.52 | **+35.01** |
| | ANIMALS10 | baseline | 32.68 | 43.20 | 37.47 | 53.24 | 47.71 | 46.21 | 37.43 | 44.26 | 40.70 | 41.23 | 48.36 | 42.95 |
| | | ours | 87.54 | 84.00 | 83.23 | 88.63 | 86.55 | 87.31 | 91.28 | 83.52 | 89.49 | 91.70 | 84.94 | **87.11** |
| | | Δ | +54.86 | +40.80 | +45.76 | +35.39 | +38.84 | +41.10 | +53.85 | +39.26 | +48.79 | +50.47 | +36.58 | **+44.15** |
| | GTSRB | baseline | 8.78 | 15.17 | 10.23 | 19.23 | 15.63 | 14.36 | 10.99 | 12.75 | 10.50 | 21.49 | 17.32 | 14.22 |
| | | ours | 69.38 | 66.69 | 60.51 | 69.33 | 73.65 | 76.34 | 69.11 | 69.03 | 67.39 | 81.41 | 68.12 | **70.09** |
| | | Δ | +60.60 | +51.52 | +50.28 | +50.10 | +58.02 | +61.98 | +58.12 | +56.28 | +56.89 | +59.92 | +50.80 | **+55.86** |
| | ImageNet20 | baseline | 19.85 | 25.37 | 12.65 | 30.69 | 29.44 | 28.34 | 15.81 | 28.47 | 21.95 | 21.74 | 24.44 | 23.52 |
| | | ours | 63.06 | 59.42 | 60.26 | 61.07 | 61.78 | 64.78 | 62.46 | 60.20 | 65.00 | 67.42 | 58.18 | **62.15** |
| | | Δ | +43.21 | +34.05 | +47.61 | +30.38 | +32.34 | +36.44 | +46.65 | +31.73 | +43.05 | +45.68 | +33.74 | **+38.63** |
| | SVHN | baseline | 6.44 | 6.26 | 6.72 | 26.38 | 12.97 | 11.88 | 9.04 | 6.55 | 19.58 | 15.44 | 19.42 | 12.79 |
| | | ours | 87.88 | 85.36 | 66.21 | 90.36 | 84.44 | 89.36 | 92.81 | 84.44 | 82.31 | 89.49 | 91.53 | **85.84** |
| | | Δ | +81.44 | +79.10 | +59.49 | +63.98 | +71.47 | +77.48 | +83.77 | +77.89 | +62.73 | +74.05 | +72.11 | **+73.05** |

**Defense.** We evaluate the effectiveness of our work using the following four adversarial training mechanisms: Gen-AF (Zhou et al., 2024), PGD-AT (Madry et al., 2018), TRADES (Zhang et al., 2019) and MART (Wang et al., 2019).

**Datasets and models.** We evaluate our method on the six datasets: CIFAR10 (Krizhevsky et al., 2009), STL10 (Coates et al., 2011), ANIMALS10 (Alessio, 2020), GTSRB (Stallkamp et al., 2012), ImageNet20 (Russakovsky et al., 2015), and SVHN (Netzer et al., 2011). We use the publicly available pre-trained encoders from *solo-learn*[2], a well-established SSL library, as victim encoders. Following (Zhou et al., 2023; 2024), we select eleven SSL methods (W-MSE (Ermolov et al., 2021), BYOL (Grill et al., 2020), NNCLR (Dwibedi et al., 2021), SimCLR (Chen et al., 2020a), MoCo v2+ (Chen et al., 2020b), MoCo v3 (Chen et al., 2021), SupCon (Khosla et al., 2020), RESSL (Zheng et al., 2021), DINO (Caron et al., 2021), SwAV (Caron et al., 2020), and VibCreg (Lee & Aune, 2021)). All encoders are pre-trained on either CIFAR10 (Krizhevsky et al., 2009) or ImageNet (Russakovsky et al., 2015), using ResNet18 as the backbone.

**Evaluation Metrics**. To evaluate the effectiveness of our proposed method, we examine the model's robustness and generalization capability using the following three metrics.

**Benign Test Accuracy:** Benign Test Accuracy (BA) denotes the classification accuracy on clean samples. Higher values indicate better generalization ability.

**Robust Test Accuracy:** Robust Test Accuracy (RA) measures the classification accuracy on adversarial examples. Higher values signify greater robustness.

**Attack Success Rate:** Attack Success Rate (ASR) is a metric from the attacker's perspective, which provides insights into model robustness. It represents the proportion of adversarial examples where the model's prediction differs from that of the corresponding clean sample. Higher values reflect stronger attack performance.

**Implementation details.** Adversarial tuning is performed for 20 epochs with a batch size of 256, optimized by the Adam optimizer (Kingma, 2014). The classification head is composed of three fully connected layers. Following the previous work (Zhou et al., 2023), we select CIFAR10 as the

---

[2]https://github.com/vturrisi/solo-learn

Table 3: BA(%) comparison between the baseline method and ZePAD in the semi-black-box setting, where $\mathcal{D}_f$ and $\mathcal{D}_d$ denote the adversarial training and downstream datasets, respectively.

| $\mathcal{D}_f$ | $\mathcal{D}_d$ | Method | W-MSE | BYOL | NNCLR | SimCLR | MoCo2+ | MoCo3 | SupCon | RESSL | DINO | SwAV | VibCreg | AVG |
|---|---|---|---|---|---|---|---|---|---|---|---|---|---|---|
| CIFAR10 | STL10 | baseline | 78.49 | 83.08 | 83.09 | 81.94 | 84.46 | 84.00 | 85.29 | 81.87 | 83.54 | 81.53 | 81.64 | 82.63 |
| | | ours | 84.84 | 84.88 | 85.91 | 86.52 | 85.34 | 85.21 | 82.93 | 84.23 | 84.88 | 84.56 | 84.11 | **84.86** |
| | | Δ | +6.35 | +1.80 | +2.82 | +4.58 | +0.88 | +1.21 | -2.36 | +2.36 | +1.34 | +3.03 | +2.47 | **+2.23** |
| | ANIMALS10 | baseline | 80.39 | 90.22 | 91.22 | 90.82 | 88.03 | 88.05 | 88.23 | 87.86 | 85.01 | 89.16 | 94.70 | 88.52 |
| | | ours | 91.31 | 92.37 | 93.18 | 92.43 | 92.01 | 92.64 | 91.52 | 91.43 | 91.66 | 92.00 | 92.92 | **92.13** |
| | | Δ | +10.92 | +2.15 | +1.96 | +1.61 | +3.98 | +4.59 | +3.29 | +3.57 | +6.65 | +2.84 | -1.78 | **+3.62** |
| | GTSRB | baseline | 73.76 | 75.99 | 79.06 | 78.61 | 76.37 | 81.11 | 81.65 | 76.03 | 80.63 | 85.51 | 82.25 | 79.18 |
| | | ours | 82.50 | 83.17 | 83.86 | 84.36 | 83.10 | 84.49 | 82.13 | 82.71 | 82.25 | 84.33 | 83.41 | **83.30** |
| | | Δ | +8.74 | +7.18 | +4.80 | +5.75 | +6.73 | +3.38 | +0.48 | +6.68 | +1.62 | -1.18 | +1.16 | **+4.12** |
| | ImageNet20 | baseline | 59.01 | 67.47 | 65.94 | 57.74 | 66.22 | 65.00 | 58.87 | 61.95 | 66.45 | 63.01 | 62.31 | 63.09 |
| | | ours | 64.89 | 66.23 | 66.86 | 66.10 | 66.21 | 66.49 | 62.56 | 64.47 | 66.63 | 62.75 | 62.75 | **65.33** |
| | | Δ | +5.88 | -1.24 | +0.92 | +8.36 | -0.01 | +1.49 | +3.69 | +2.52 | -1.03 | +3.62 | +0.44 | **+2.24** |
| | SVHN | baseline | 57.92 | 58.59 | 66.43 | 64.81 | 64.67 | 64.32 | 69.88 | 67.65 | 70.38 | 72.57 | 73.26 | 66.41 |
| | | ours | 75.12 | 78.62 | 80.31 | 79.43 | 73.14 | 77.73 | 75.39 | 77.55 | 77.87 | 75.88 | 77.32 | **77.12** |
| | | Δ | +17.20 | +20.03 | +13.88 | +14.62 | +8.47 | +13.41 | +5.51 | +9.90 | +7.49 | +3.31 | +4.06 | **+10.72** |
| GTSRB | CIFAR10 | baseline | 56.52 | 68.60 | 72.38 | 70.56 | 70.35 | 72.37 | 71.35 | 71.20 | 71.75 | 68.46 | 73.35 | 69.72 |
| | | ours | 91.20 | 91.07 | 90.92 | 94.40 | 91.54 | 91.24 | 91.11 | 90.91 | 91.06 | 91.53 | 90.65 | **91.42** |
| | | Δ | +34.68 | +22.47 | +18.54 | +23.84 | +21.19 | +18.87 | +19.76 | +19.71 | +19.31 | +23.07 | +17.30 | **+21.70** |
| | STL10 | baseline | 51.99 | 63.37 | 65.41 | 66.14 | 64.78 | 63.21 | 64.79 | 64.89 | 64.37 | 62.90 | 68.76 | 63.69 |
| | | ours | 81.19 | 80.38 | 80.64 | 82.98 | 80.80 | 81.19 | 80.24 | 80.68 | 80.79 | 80.91 | 79.05 | **80.80** |
| | | Δ | +29.20 | +17.01 | +15.23 | +16.84 | +16.02 | +17.98 | +15.45 | +15.79 | +16.42 | +18.01 | +10.29 | **+17.11** |
| | ANIMALS10 | baseline | 48.53 | 63.82 | 68.81 | 70.69 | 65.86 | 68.31 | 74.65 | 64.75 | 62.19 | 61.75 | 76.57 | 65.99 |
| | | ours | 89.75 | 90.07 | 90.29 | 89.05 | 89.66 | 90.69 | 89.79 | 90.11 | 89.10 | 89.80 | 89.63 | **89.81** |
| | | Δ | +41.22 | +26.25 | +21.48 | +18.36 | +23.80 | +22.38 | +15.14 | +25.36 | +26.91 | +28.05 | +13.06 | **+23.82** |
| | ImageNet20 | baseline | 35.64 | 48.76 | 56.18 | 56.44 | 52.35 | 54.36 | 47.40 | 51.49 | 51.13 | 47.84 | 54.06 | 50.51 |
| | | ours | 61.61 | 60.66 | 60.09 | 59.96 | 59.52 | 60.71 | 60.32 | 59.87 | 60.00 | 61.17 | 57.41 | **60.12** |
| | | Δ | +25.97 | +11.90 | +3.91 | +3.52 | +7.17 | +6.35 | +12.92 | +8.38 | +8.87 | +13.33 | +3.35 | **+9.61** |
| | SVHN | baseline | 50.45 | 62.86 | 68.40 | 69.25 | 62.50 | 64.31 | 67.40 | 63.95 | 71.31 | 66.72 | 69.51 | 65.15 |
| | | ours | 81.67 | 80.36 | 82.15 | 82.42 | 80.87 | 82.96 | 79.53 | 81.19 | 82.87 | 81.80 | 82.29 | **81.65** |
| | | Δ | +31.22 | +17.50 | +13.75 | +13.17 | +18.37 | +18.65 | +12.13 | +17.24 | +11.56 | +15.08 | +12.78 | **+16.50** |

dataset for the attacker and set the perturbation limit to $10/255$. The other experimental settings can be found in *Appendix E*.

The default experimental setting follows a semi-black-box scenario. When the victim encoder in the MPAE-Branch is not BYOL, we select BYOL as the adversarial auxiliary encoder for the MPAE-Branch. Conversely, if the victim encoder is BYOL, we choose NNCLR as the adversarial auxiliary encoder (AdvAu-Model). The encoder in the BMP-Branch is trained using SimCLR with local datasets. Besides, we also validate our method under the white-box scenario, and the experimental results can be found in *Appendix F*.

## 5.2 SEMI-BLACK-BOX SCENARIO

In this scenario, the attacker knows the SSL method employed by the victim model, as well as the pre-training dataset. We evaluate the performance of ZePAD across 11 SSL methods and 6 datasets, selecting AdvEncoder as the attack method due to its status as a SOTA downstream-agnostic adversarial examples method. This method enables effective attacks without knowledge of the pre-trained dataset or downstream tasks.

We present comparisons between ZePAD and the baseline in Tables 1 and 2, where the adversarial training dataset and downstream task dataset are identical. The results show that ZePAD significantly enhances task performance on benign samples, with average accuracy improvements of 29.20% and 32.03% on the SVHN and ANIMALS10 datasets, respectively, when the pre-training dataset is ImageNet. In contrast, the baseline's accuracy drops significantly on adversarial examples, with RA falling below 10% in some experimental settings, underscoring the threat posed by AdvEncoder-generated DAEs to pre-trained encoders. As shown in Table 4, ZePAD significantly improves RA. For instance, when the pre-training dataset is CIFAR10 and the downstream task is SVHN, RA increases by 73.86% compared to the baseline, reaching an average RA of 86.97%. This demonstrates that ZePAD

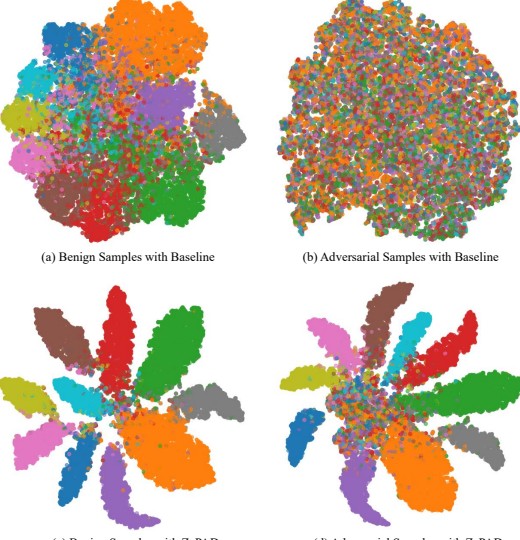

(a) Benign Samples with Baseline  (b) Adversarial Samples with Baseline

(c) Benign Samples with ZePAD  (d) Adversarial Samples with ZePAD

Figure 3: t-SNE visualization of feature spaces for the baseline and ZePAD under benign and adversarial examples. Different colors indicate different classes.

Table 4: RA (%) comparison between the baseline method and ZePAD in the semi-black-box setting, where $\mathcal{D}_f$ and $\mathcal{D}_d$ denote the adversarial training and downstream datasets, respectively.

| $\mathcal{D}_f$ | $\mathcal{D}_d$ | Method | W-MSE | BYOL | NNCLR | SimCLR | MoCo2+ | MoCo3 | SupCon | RESSL | DINO | SwAV | VibCreg | AVG |
|---|---|---|---|---|---|---|---|---|---|---|---|---|---|---|
| CIFAR10 | STL10 | baseline | 41.16 | 46.85 | 44.05 | 67.52 | 65.55 | 64.28 | 56.89 | 51.69 | 31.77 | 59.55 | 43.07 | 52.03 |
| | | ours | 75.60 | 73.28 | 72.84 | 77.12 | 70.54 | 72.45 | 70.76 | 73.99 | 73.67 | 74.04 | 70.83 | 73.19 |
| | | Δ | +34.44 | +26.43 | +28.79 | +9.60 | +4.99 | +8.17 | +13.87 | +22.30 | +41.90 | +14.49 | +27.76 | +21.16 |
| | ANIMALS10 | baseline | 48.31 | 49.25 | 49.23 | 63.48 | 63.03 | 60.32 | 58.48 | 54.92 | 35.46 | 63.75 | 50.19 | 54.22 |
| | | ours | 75.49 | 72.15 | 74.21 | 75.23 | 69.35 | 70.76 | 74.60 | 77.62 | 74.71 | 73.33 | 75.90 | 73.94 |
| | | Δ | +27.18 | +22.90 | +24.98 | +11.75 | +6.32 | +10.44 | +16.12 | +22.70 | +39.25 | +9.58 | +25.71 | +19.72 |
| | GTSRB | baseline | 10.85 | 12.41 | 11.29 | 24.63 | 27.02 | 27.60 | 13.22 | 12.37 | 10.62 | 27.82 | 11.56 | 17.22 |
| | | ours | 60.50 | 56.16 | 53.71 | 54.24 | 48.10 | 52.58 | 53.83 | 59.50 | 56.48 | 54.03 | 55.87 | 55.00 |
| | | Δ | +49.65 | +43.75 | +42.42 | +29.61 | +21.08 | +24.98 | +40.61 | +47.13 | +45.86 | +26.21 | +44.31 | +37.78 |
| | ImageNet20 | baseline | 27.60 | 34.41 | 29.82 | 38.24 | 42.87 | 41.60 | 28.97 | 35.10 | 21.24 | 37.69 | 27.13 | 33.15 |
| | | ours | 48.78 | 53.97 | 55.15 | 52.18 | 50.16 | 49.40 | 50.98 | 52.97 | 52.05 | 52.71 | 46.34 | 51.34 |
| | | Δ | +21.18 | +19.56 | +25.33 | +13.94 | +7.29 | +7.80 | +22.01 | +17.87 | +30.81 | +15.02 | +19.21 | +18.18 |
| | SVHN | baseline | 9.16 | 11.68 | 14.99 | 14.58 | 21.51 | 16.38 | 11.10 | 9.27 | 8.95 | 16.08 | 10.55 | 13.11 |
| | | ours | 55.84 | 61.37 | 52.97 | 64.75 | 50.86 | 59.39 | 62.74 | 68.26 | 64.37 | 58.24 | 55.34 | 59.47 |
| | | Δ | +46.68 | +49.69 | +37.98 | +50.17 | +29.35 | +43.01 | +51.64 | +58.99 | +55.42 | +42.16 | +44.79 | +46.35 |
| GTSRB | CIFAR10 | baseline | 11.78 | 11.78 | 12.80 | 24.31 | 16.49 | 12.66 | 14.00 | 11.33 | 15.80 | 12.44 | 20.42 | 14.89 |
| | | ours | 53.62 | 62.57 | 64.43 | 60.52 | 63.91 | 67.67 | 60.76 | 61.94 | 64.32 | 65.74 | 63.80 | 62.66 |
| | | Δ | +41.84 | +50.79 | +51.63 | +36.21 | +47.42 | +55.01 | +46.76 | +50.61 | +48.52 | +53.30 | +43.38 | +47.77 |
| | STL10 | baseline | 26.76 | 33.02 | 22.42 | 46.23 | 40.43 | 42.37 | 25.12 | 41.98 | 33.68 | 42.77 | 40.84 | 35.97 |
| | | ours | 57.90 | 65.88 | 67.69 | 73.11 | 68.58 | 69.58 | 66.16 | 66.77 | 68.18 | 68.13 | 65.32 | 67.03 |
| | | Δ | +31.14 | +32.86 | +45.27 | +26.88 | +28.15 | +27.21 | +41.04 | +24.79 | +34.50 | +25.36 | +24.48 | +31.06 |
| | ANIMALS10 | baseline | 32.68 | 43.20 | 37.47 | 53.24 | 47.71 | 46.21 | 37.43 | 44.26 | 40.70 | 41.23 | 48.36 | 42.95 |
| | | ours | 65.49 | 70.52 | 72.78 | 72.67 | 70.85 | 71.77 | 71.65 | 73.86 | 71.47 | 72.37 | 71.96 | 71.40 |
| | | Δ | +32.81 | +27.32 | +35.31 | +19.43 | +23.14 | +25.56 | +34.22 | +29.60 | +30.77 | +31.14 | +23.60 | +28.45 |
| | ImageNet20 | baseline | 19.85 | 25.37 | 12.65 | 30.69 | 29.44 | 28.34 | 15.81 | 28.47 | 21.95 | 21.74 | 24.44 | 23.52 |
| | | ours | 39.44 | 45.67 | 48.10 | 48.81 | 47.53 | 48.83 | 46.13 | 46.30 | 48.30 | 47.61 | 43.83 | 46.41 |
| | | Δ | +19.59 | +20.30 | +35.45 | +18.12 | +18.09 | +20.49 | +30.32 | +17.83 | +26.35 | +25.87 | +19.39 | +22.89 |
| | SVHN | baseline | 6.44 | 6.26 | 6.72 | 26.38 | 12.97 | 11.88 | 9.04 | 6.55 | 19.58 | 15.44 | 19.42 | 12.79 |
| | | ours | 55.72 | 68.17 | 71.15 | 63.38 | 65.98 | 67.89 | 62.79 | 69.54 | 74.75 | 63.19 | 57.09 | 65.42 |
| | | Δ | +49.28 | +61.91 | +64.43 | +37.00 | +53.01 | +56.01 | +53.75 | +62.99 | +55.17 | +47.75 | +37.67 | +52.63 |

can effectively enhance model robustness while improving generalization, achieving ***Zero-Sacrifice Adversarial Defense***.

To investigate ZePAD's cross-task defense capabilities, we design experiments where the adversarial training dataset differs from the downstream task dataset, with CIFAR10 uniformly selected as the pre-training dataset. The experimental results are provided in Tables 3 and 4. The results demonstrate that ZePAD can still enhance BA compared to baseline, even when the adversarial training dataset and downstream dataset are mismatched. For example, when the fine-tuning dataset is GTSRB, BA improves by an average of 21.70% and 23.82% on CIFAR10 and ANIMAL10 downstream tasks, respectively. Against adversarial examples, ZePAD continues to perform well, significantly boosting RA. When fine-tuning on CIFAR10, RA averages 73.19% on STL10 and 73.94% on ANIMALS10, showing robust defense. This confirms ZePAD's effectiveness across tasks, achieving ***Persistent-Robustness*** adversarial defense.

To intuitively understand ZePAD's impact on model generalization and robustness, we conducted t-SNE visualization when the pre-trained dataset was CIFAR10 and the downstream task was SVHN, with results shown in Figure 3. It is clear that ZePAD sharpens classification boundaries on benign samples. When facing adversarial exsamples, the baseline method almost loses classification capability, but ZePAD can still correctly classify samples.

## 5.3 ABLATION STUDY

In this section, we investigate the impact of different components of ZePAD. We use a pre-trained DINO encoder on CIFAR10 and conduct experiments on ANIMALS10.

We analyze the effects of adversarial training, the MPAD-Branch, and the BMP-Branch on the overall method, with results in Table 5. Removing adversarial training signifi-

Table 5: The ablation study of the proposed ZePAD.

| Adversarial Fine-Tuning | MPAD Branch | BMP Branch | BA | RA | ASR |
|---|---|---|---|---|---|
| × | × | × | 85.01 | 35.46 | 63.35 |
| × | ✓ | ✓ | 90.84 | 59.40 | 38.89 |
| ✓ | × | ✓ | 91.57 | 70.35 | 26.71 |
| ✓ | ✓ | × | 84.51 | 72.47 | **19.36** |
| ✓ | ✓ | ✓ | **91.66** | **74.71** | 21.99 |

cantly reduces RA and increases ASR, indicating its effectiveness in enhancing robustness. Nonetheless, BA and RA still improve compared to standard testing, showing that pre-trained encoders from different self-supervised methods have complementary feature extraction patterns. When MPAD-Branch is removed, RA decreases with little change in BA, highlighting its role in boosting robustness without harming generalization. If BMP-Branch is removed, BA drops below standard testing levels, indicating that adversarial training can negatively affect generalization. Thus, BMP-Branch is crucial for maintaining and enhancing the model's generalization capability. The other ablation studies can be found in ***Appendix G***.

Table 6: DAE sample detection accuracy(%), where $\mathcal{D}_f$ and $\mathcal{D}_d$ denote the adversarial training and downstream datasets.

| $\mathcal{D}_f$ and $\mathcal{D}_d$ | W-MSE | BYOL | NNCLR | SimCLR | MoCo2+ | MoCo3 | SupCon | RESSL | DINO | SwAV | VibCreg | AVG |
|---|---|---|---|---|---|---|---|---|---|---|---|---|
| CIFAR10 | 86.20 | 83.71 | 83.46 | 86.18 | 81.53 | 82.81 | 84.27 | 83.06 | 70.97 | 81.50 | 84.01 | 82.52 |
| STL10 | 72.96 | 76.41 | 75.05 | 79.65 | 72.38 | 75.96 | 78.93 | 73.59 | 59.75 | 72.98 | 79.43 | 74.28 |
| ANIMALS10 | 80.91 | 81.55 | 80.31 | 81.98 | 74.57 | 76.66 | 83.32 | 81.46 | 71.02 | 73.11 | 84.55 | 79.04 |
| GTSRB | 73.65 | 75.03 | 71.67 | 76.96 | 73.93 | 73.15 | 75.69 | 70.29 | 56.09 | 69.96 | 76.62 | 72.09 |
| ImageNet20 | 72.25 | 77.99 | 76.65 | 81.23 | 73.49 | 76.48 | 82.67 | 77.71 | 64.08 | 72.21 | 83.09 | 76.17 |
| SVHN | 59.26 | 64.93 | 63.82 | 70.75 | 62.17 | 66.69 | 67.75 | 62.69 | 52.38 | 64.07 | 68.45 | 63.91 |

## 5.4 DETECTING DAEs BY CONFIDENCE SCORES

We find that the significant confidence score differences between the MPAE and BMP branches for benign and adversarial exsamples, as illustrated in Figure 2. Then, we are curious that if we can utilize the inherent sensitivity to detect DAEs directly without additional DAE identification training.

Specifically, we define a sample as a DAE if the confidence of the MPAE-Branch exceeds that of the BMP-Branch by a threshold of 0.1. The adversarial exsamples in this experiment were generated from AdvEncoder (Zhou et al., 2023). The results in Table 6 demonstrate high effectiveness. We achieve an average accuracy of 82.52% when the fine-tuning and downstream datasets are consistent (CIFAR10) and 79.04% when they are not (ANIMALS10). The higher accuracy with consistent datasets is attributed to smoother confidence score distributions, which allows for a clearer distinction between benign and adversarial inputs. Additionally, to further discuss the extracted patterns from different branches for various inputs, the visualization can be found in *Appendix H*.

## 5.5 COMPARISON STUDY

In this section, we compare the proposed ZePAD with other advanced adversarial defense methods: Gen-AF, TRADES, MART, and PGD-AT. We use ImageNet for pre-training and STL10 as the downstream dataset. As shown in Table 7, ZePAD outperforms existing methods in both robustness and generalization.

Table 7: Comparison study(%).

| Model | Method | BA | UAP RA | UAPGD RA | SSP RA | PAP RA | AdvEncoder RA |
|---|---|---|---|---|---|---|---|
| SimCLR | PGD-AT(2018 ICLR) | 22.19 | 21.23 | 23.00 | 21.79 | 20.97 | 19.04 |
| | MART(2019 ICLR) | 42.79 | 42.50 | 42.65 | 42.60 | 42.24 | 42.36 |
| | TRADES(2019 PMLR) | 50.79 | 50.40 | 50.51 | 50.95 | 50.59 | 50.66 |
| | Gen-AF(2024 S&P) | 68.69 | 64.43 | 68.65 | 65.92 | 65.48 | 62.28 |
| | ZePAD(ours) | **82.07** | **77.56** | **76.51** | **72.93** | **72.10** | **73.26** |
| | Δ | **+13.38** | **+13.13** | **+7.86** | **+7.01** | **+6.62** | **+10.98** |
| MoCo v2+ | PGD-AT(2018 ICLR) | 13.72 | 12.15 | 13.48 | 12.30 | 13.02 | 12.70 |
| | MART(2019 ICLR) | 42.79 | 41.46 | 41.52 | 41.28 | 41.76 | 41.36 |
| | TRADES(2019 PMLR) | 50.50 | 52.40 | 50.60 | 52.37 | 52.60 | 52.28 |
| | Gen-AF(2024 S&P) | 69.09 | 64.98 | 68.15 | 59.41 | 60.47 | 66.00 |
| | ZePAD(ours) | **81.60** | **76.05** | **74.83** | **72.15** | **67.96** | **74.30** |
| | Δ | **+12.51** | **+11.07** | **+6.68** | **+12.74** | **+7.49** | **+8.30** |

"Δ" represents the improvement ratio compared to the second-best metric.

tion. ZePAD's BA remains above 80%, indicating its effectiveness in boosting model generalization. In terms of robustness, ZePAD achieves over 70% RA against six attacks, outperforming all other methods and further demonstrating its zero-sacrifice characteristic. Besides, the computational cost comparison can be found in *Appendix I*. Finally, we extend our defense method to other tasks, and the corresponding discussions and experiments are provided in *Appendix J*.

## 6 CONCLUSION

In this paper, we introduced ZePAD, the first zero-sacrifice persistent-robustness adversarial defense method designed for pre-trained encoders. The core idea behind ZePAD is to enhance adversarial robustness and benign performance by simultaneously leveraging two complementary branches. Our experimental results across various SSL methods and multiple datasets demonstrate that ZePAD provides robust performance in different settings, effectively enhancing adversarial robustness without sacrificing benign performance. Additionally, ZePAD's persistent-robustness capability enables it to adapt to new tasks with a single tuning, making it scalable and adaptable to real-world applications. Further investigation is required in several areas. One important direction for future work is improving the interpretability of the ZePAD framework and DAEs. Gaining insights into them could help refine the model and offer more transparency.

## 7 ETHICS STATEMENT

This work adheres to the ICLR Code of Ethics. In this study, no human subjects or animal experimentation was involved. All datasets used, including CIFAR10, STL10, ANIMALS10, SVHN, ImageNet and GTSRB were sourced in compliance with relevant usage guidelines, ensuring no violation of privacy. We have taken care to avoid any biases or discriminatory outcomes in our research process. No personally identifiable information was used, and no experiments were conducted that could raise privacy or security concerns. We are committed to maintaining transparency and integrity throughout the research process.

## 8 REPRODUCIBILITY STATEMENT

We have made every effort to ensure that the results presented in this paper are reproducible. Specifically, our method is detailed in ***Section 4***, and the main steps are provided in ***Appendix D*** to facilitate reproducibility. Moreover, all code will be made publicly available after the review cycle to ensure the reproducibility. Details of the experimental settings, including datasets, models, SSL methods, training configurations, and hyperparameter choices, are presented in ***Section 5*** and ***Appendix E***. All datasets used in our experiments are publicly available, and the download links for all open-source pre-trained models involved in the experiments are released together with our code.

## 9 ACKNOWLEDGMENTS

This work was supported in part by the National Natural Science Foundation of China under Grant 62271335, in part by the Sichuan Science and Technology Program under Grant 2025ZNSFSC0470 and in part by Tianfu Jiangxi Laboratory.

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

# Appendix of Paper:
## "*Zero-Sacrifice Persistent-Robustness Adversarial Defense for Pre-Trained Encoders*"

This appendix provides key information to help enhance the understanding of our ZePAD. To help readers better understand the context of this field, we first provide the related works in Appendix A. Then, Appendix B covers the preliminaries, including the threat model, the defender's objectives, and the associated challenges. We give a through theoretical proof to support our proposed method in Appendix C. Then, in Appendix D, we outline the main steps of the proposed ZePAD. To help readers reproduce our proposed method, we give the detailed experimental settings in Appendix E. Additional experiments in white-box scenarios are presented in Appendix F. Further ablation studies are illustrated in Appendix G. The extracted patterns from different SSL methods are visualized in Appendix H. Finally, discussions regarding resource occupancy and consumption can be found in Appendix I.

## A    RELATED WORKS

### A.1    SELF-SUPERVISED LEARNING

SSL is an emerging machine learning paradigm that leverages vast amounts of unlabeled data to pre-train a generic encoder, which can be applied to various downstream tasks, such as image classification and object detection. This approach overcomes the limitations of labeled data and achieves robust feature extraction, enabling users with limited resources to utilize pre-trained encoders without training from scratch. Fine-tuning, which is generally lightweight, is the primary method used to adapt these encoders to specific tasks.

Fine-tuning a pre-trained encoder enhances its adaptability to downstream tasks. Common fine-tuning methods include linear evaluation (Aghajanyan et al., 2020; Kumar et al., 2022), which involves adjusting only the last layer, and full fine-tuning (Aghajanyan et al., 2020; Kumar et al., 2022; Hendrycks et al., 2019; Miller et al., 2021), which involves tuning all layers of the encoder.

Existing self-supervised learning approaches can be broadly categorized into several types (Fini et al., 2022; Tao et al., 2022). Contrastive learning methods, such as MoCo (Chen et al., 2020b; 2021) and SimCLR (Chen et al., 2020a), train models by ensuring that feature vectors of similar samples are close, while those of dissimilar samples are distant. Negative-free methods, such as BYOL (Grill et al., 2020) and SimSiam (Chen & He, 2021) improve representation by maintaining consistency between positive samples without the use of negative samples. Clustering-based methods, including SwAV (Caron et al., 2020), employ traditional clustering techniques to group similar samples into the same category. Redundancy reduction-based methods, such as W-MSE (Ermolov et al., 2021) and VIbCReg (Lee & Aune, 2021), enhance the quality of representations by strengthening connections within the same dimension and attempting to decouple across different dimensions.

### A.2    ADVERSARIAL ATTACKS AND DEFENSES.

Recent studies (Goodfellow et al., 2014; Hu et al., 2022) have revealed the vulnerability of deep neural networks (DNNs) to adversarial examples. Universal adversarial perturbations (UAPs) (Co et al., 2019) possess attack universality, as a single perturbation can affect multiple samples. Adversarial attacks can be categorized into white-box (Naseer et al., 2020) and black-box attacks (Lin et al., 2024) scenarios. In white-box scenarios, the attacker can access the model's internal structure and parameters, while in black-box scenarios, only the model's inputs and outputs are observable. With the increasing use of pre-trained encoders, recent research (Ban & Dong, 2022; Zhou et al., 2023; Zhang et al., 2022) has begun to explore attacks targeting these encoders. These attacks are downstream-agnostic, affecting any downstream model that uses the pre-trained encoder, thus highlighting the security risks inherent in the pre-trained paradigm.

Existing adversarial defense methods mainly fall into two categories: input filtering (Guo et al., 2017) and enhancing the model's inherent robustness (Zhu & Gupta, 2017; Zhou et al., 2024), the latter of which often employs adversarial training. While adversarial training can strengthen

a model's robustness, it may also affect its generalization ability, leading to decreased performance on clean samples. Recent research (Lamb et al., 2019; Zhou et al., 2024; Liu et al., 2021; Li et al., 2024) has focused on balancing the enhancement of model robustness with the preservation of generalization capabilities. However, existing methods have yet to achieve the enhancement of model robustness without compromising, or even while improving, its generalization ability.

## B  PRELIMINARIES

### B.1  THREAT MODEL

This paper primarily evaluates the security and vulnerability of pre-trained encoders in the context of downstream classification tasks. To begin our analysis, we first define the attacker's goals and the defender's capabilities.

**Attacker's Goals:** Building on previous works Ban & Dong (2022); Zhou et al. (2023; 2024), we assume that attackers are likely to exploit publicly available pre-trained encoders, which can be easily accessed through purchase or directly downloaded from open platforms. The primary goal of these attackers is to create adversarial examples targeting downstream models that utilize these pre-trained encoders, with the intention of launching non-targeted attacks. Due to their limited knowledge of downstream tasks, these attacks exhibit the following key characteristics:

- **Universality.** Adversarial examples can successfully attack any downstream model using the pre-trained encoder, regardless of the specific downstream task.
- **Effectiveness.** These adversarial examples can compromise the performance of downstream models, even after fine-tuning and the application of defense techniques.
- **Stealthiness.** The adversarial perturbations added to normal examples must remain imperceptible to humans, ensuring they go undetected.

**Defender's Capabilities:** Defenders do not control the pre-training procedure and lack direct knowledge of it, including the self-supervised learning method and the dataset used. However, they have full access to publicly available pre-trained encoders and complete control over the fine-tuning process. This includes the ability to modify the structure of the downstream model as needed.

### B.2  DEFENDER'S GOAL

Our method aims to leverage the powerful feature representations of pre-trained encoders to enhance downstream task performance. Specifically, we aim to accomplish the following objectives:

- **Zero-Sacrifice Capability.** This goal ensures that models built on pre-trained encoders maintain high classification accuracy on benign samples without compromising performance. Ideally, it also seeks to improve performance on benign data.
- **Adversarial Robustness.** This goal enables downstream models to effectively mitigate adversarial examples, particularly those generated by pre-trained encoders. It enhances the model's robustness, ensuring correct predictions even when the input is adversarially perturbed.
- **Resource Efficiency:** This goal ensures that the defense method remains resource-efficient, requiring no additional external data or computational resources beyond those available for the downstream task, making it practical and scalable for real-world applications.
- **Persistent-Robustness Capability.** This goal ensures that our method provides robust performance across a wide range of downstream tasks with a single tuning. It enables persistent-robustness adaptability, allowing the model to perform effectively as new tasks emerge, without requiring frequent retraining or task-specific tuning.

### B.3  CHALLENGES

In this paper, we propose challenging but crucial goals. Unlike previous works, we aim to enhance adversarial robustness without sacrificing benign performance, and even expect to improve it.

Furthermore, we aim to establish a persistent-robustness paradigm that enables a one-time tuning process while ensuring robustness across all downstream tasks.

We aim to leverage the inherent characteristics of neural networks to develop a novel defense paradigm. However, re-tuning the pre-trained encoder is not an option, as any inappropriate alterations may compromise its powerful feature extraction capabilities. These constraints make our goals particularly challenging, and the main challenges are outlined below:

**Challenge I: The Challenge of Zero-Sacrifice.** Our approach significantly differs from previous methods, particularly in how it addresses the trade-off between adversarial robustness and benign performance. Traditional methods aim to minimize performance loss on benign samples to avoid a drastic decline in accuracy. However, these methods are inherently limited by the adversarial fine-tuning paradigm, where fine-tuning is performed based on adversarial examples after pre-training the encoder. This creates a conflict between optimizing adversarial robustness and maintaining high performance on clean data. Beyond this, this paper proposes a more challenging goal: achieving the zero-sacrifice characteristic, where both adversarial robustness and benign performance are optimized simultaneously. This is more challenging because defenders have no access to the pre-training process. Furthermore, the optimization goals of benign pre-training and adversarial fine-tuning are inherently in conflict, making it difficult to optimize both adversarial robustness and benign performance simultaneously.

**Challenge II: The Challenge of Persistent-Robustness Capability.** In addition to setting a higher goal for benign sample performance, we introduce the novel capability of "***Persistent-Robustness Capability***", which has not been addressed in previous approaches. As mentioned earlier, previous methods require task-specific tuning for each downstream task. This introduces significant overhead, as the network must be retrained whenever the downstream task changes, increasing training costs and computational demands. To address this, we introduce the concept of persistent-robustness capability, which ensures that the model can adapt to new tasks without frequent retraining. This is a challenging goal because it requires the defense method to be both upstream- and downstream-agnostic. In other words, the method must adapt to various pre-training procedures and downstream tasks without relying on task-specific or model-specific assumptions, adding significant complexity to its design. Although this goal is crucial, it has often been overlooked in previous works. Achieving it will not only allow the defense method to support a broader range of applications but also provide valuable insights for developing more explainable and transparent defense mechanisms.

# C  THEORETICAL PROOF

The key rationale of ZePAD is that neural networks naturally assign higher confidence to inputs that follow their training data distribution, and lower confidence to inputs that lie outside that distribution. This behavior is not manually designed, but emerges naturally from empirical risk minimization using cross-entropy. To support this statement, we present a theoretical proof as follows:

## C.1  TRAINING DISTRIBUTIONS OF TWO BRANCHES

Training Distributions of Two Branches Let $Q(x, y)$ denote the benign data distribution, where $x$ is the input image and $y \in \{1, \ldots, K\}$ is the label. Then, $Q_\delta(x, y)$ represents the adversarial data distribution. Each sample has the form $x + \delta$ with $\|\delta\|_p \leq \varepsilon$, where $\delta$ is the adversarial perturbation vector crafted by an attacker

Each branch is optimized on its own empirical training distribution, denoted as $P_b$ and $P_a$, respectively. In particular, the MPAE branch is trained on a mixture of benign and adversarial examples. Hence, $P_b$ and $P_a$ can be formulated as:

$$P_b(x, y) = Q(x, y), \tag{S1}$$

$$P_a(x, y) = (1 - \alpha)\, Q(x, y) + \alpha\, Q_\delta(x, y), \quad \alpha \in (0, 1), \tag{S2}$$

where $\alpha$ denotes the mixing coefficient that controls the proportion of adversarial examples used during training.

Formally, $P_b$ serves as the empirical training distribution for the BMP branch, while $P_a$ represents the mixed training distribution for the MPAE branch, containing both benign and adversarial examples.

### C.2 Cross-Entropy Training and Posterior Approximation

Consider a classifier $f(x)$ trained using the cross-entropy loss:

$$\mathcal{L}(f) = \mathbb{E}_{(x,y)\sim P}[-\log f_y(x)], \tag{S3}$$

where $f_y(x)$ denotes the predicted probability for class $y$.

Under sufficient model capacity and training convergence, the classifier trained on distribution $P(x, y)$ approximates the true posterior Zhang (2004), which can be formulated as $f^{\star}(x) \approx P(y \mid x)$, where $f^{\star}$ refers to the theoretically optimal classifier. For our two branches, we have $f_b(x) \approx P_b(y|x)$ and $f_a(x) \approx P_a(y|x)$, where $f_b(x)$ and $f_a(x)$ denote the output probability vectors of the BMP branch and the MPAE branch, respectively. Then, we define the model confidence using the maximum softmax probability:

$$m_b(x) = \max_k f_{b,k}(x) \approx \max_k P_b(y = k \mid x), \tag{S4}$$

$$m_a(x) = \max_k f_{a,k}(x) \approx \max_k P_a(y = k \mid x), \tag{S5}$$

where $f_{b,k}(x)$ (or $f_{a,k}(x)$) represents the predicted probability that input $x$ belongs to class $k$. $m_b$ and $m_a$ represent the confidence scores of the BMP and MPAE branches, respectively. Intuitively, $m_b(x)$ and $m_a(x)$ measure how well an input $x$ aligns with the data distributions $P_b$ and $P_a$, respectively.

### C.3 Model Sensitivity

Then, for benign samples $x \sim Q(x, y)$, BMP's posterior is $P_b(y|x) = Q(y|x)$, while MPAE's posterior is diluted by the adversarial mixture $P_a(y|x) = (1 - \alpha)Q(y|x) + \alpha Q_\delta(y|x)$. Obviously, for $x \sim Q(x, y)$, $Q(y|x) > Q_\delta(y|x)$. Hence, we have:

$$P_a(y|x) = (1 - \alpha)Q(y|x) + \alpha Q_\delta(y|x) < Q(y|x) = P_b(y|x). \tag{S6}$$

As proved above, we can obtain that for benign samples, the confidence of the BMP branch is higher than that under the confidence of the MPAE branch. Conversely, for adversarial inputs, the MPAE branch yields a higher confidence than the BMP branch. This confirms a clear confidence separation between the two branches, demonstrating the inherent sensitivity of neural networks to different data distributions. Specifically, a network tends to assign higher confidence to samples that align with its training distribution, while yielding lower confidence on unseen or out-of-distribution samples.

## D The Main Steps of ZePAD

To help readers follow ZePAD, we illustrate the main steps in Algorithm 1. In brief, ZePAD first prepares three encoders—two adversarially fine-tuned (MPAE-Branch) and one trained only on clean data (BMP-Branch)—along with their classification heads. During inference, it computes confidence scores for each branch and fuses their predictions via a robust federal decision mechanism, enabling persistent-robustness defense against DAEs without sacrificing clean accuracy.[3]

## E Experimental Setting

During adversarial training, we set the value of $\lambda$ to 20 by default. The model is trained for 20 epochs with a learning rate of 0.0001 and a batch size of 256, using the Adam optimizer. For downstream model training, we employ a simple three-layer classification head, with a learning rate of 0.005, a batch size of 256, and 20 training epochs. For the AdvEncoder (Zhou et al., 2023) attack method, we adopt the default parameter settings. The perturbation budget is set to $10/255$, and the batch size is fixed at 256. During training, the hyperparameter $\alpha$ is set to its default value of 5, and we use a learning rate of 0.0002 with the Adam optimizer and 20 training epochs. To help readers follow our manuscript, our codes will be made publicly available after the review process.

---

[3]Details can be found in our codes, which would be released after the review process.

---

**Algorithm 1** The main steps of the proposed ZePAD

---

**Input:** a training dataset $(x, y) \in \mathcal{D}_b$, pre-trained encoders $\mathcal{E}_{a,1}$ and $\mathcal{E}_{a,2}$, loss weight $\lambda$, a testing dataset $(x_t, y_t) \in \mathcal{D}_t$, a adversarial noise generator $G$
**Output:** a robust and persistent-robustness model
 1: STEP 1: Encoder Preparation and Downstream Training
 2: $\mathcal{E}_b \longleftarrow$ pre-train with $\mathcal{D}_b$ with SSL method
 3: **for** each $x \in \mathcal{D}_b$ **do**
 4:    $x^* = x + G(x)$
 5:    $\mathcal{D}_{adv} = \mathcal{D}_{adv} \cup \{x^*\}$
 6: **end for**
 7: $\mathcal{E}'_{a,1} \longleftarrow$ fine-tune $\mathcal{E}_{a,1}$ with $\mathcal{D}_{adv}$ and loss weight $\lambda$
 8: $\mathcal{E}'_{a,2} \longleftarrow$ fine-tune $\mathcal{E}_{a,2}$ with $\mathcal{D}_{adv}$ and loss weight $\lambda$
 9: Downstream Training
10: $\mathcal{H}_{a,1} \longleftarrow$ train with frozen $\mathcal{E}'_{a,1}$ and $\mathcal{D}_b$
11: $\mathcal{H}_{a,2} \longleftarrow$ train with frozen $\mathcal{E}'_{a,2}$ and $\mathcal{D}_b$
12: $\mathcal{H}_b \longleftarrow$ train with frozen $\mathcal{E}'_b$ and $\mathcal{D}_b$
13: STEP 2: Robust Federal Decision
14: **for** each $x_t \in \mathcal{D}_t$ **do**
15:    $y_{a,1} = \mathcal{H}_{a,1}(\mathcal{E}'_{a,1}(x_t))$
16:    $y_{a,2} = \mathcal{H}_{a,2}(\mathcal{E}'_{a,2}(x_t))$
17:    $y_b = \mathcal{H}_b(\mathcal{E}_b(x_t))$
18:    $\mathcal{W}_{a,1}, \mathcal{W}_{a,2}, \mathcal{W}_b \longleftarrow$ Based on Eq. (7)
19:    $y_{predict} = \mathcal{W}_{a,1}y_{a,1} + \mathcal{W}_{a,2}y_{a,2} + \mathcal{W}_b y_b$
20: **end for**

---

# F  EXPERIMENTS UNDER THE WHITE-BOX SCENARIO

In the white-box scenario, compared to the semi-black-box scenario, the attacker has access not only to the victim encoder but also to an additional open-source encoder chosen by the defender. In this scenario, the attacker can access all models used in the MPAE-Branch, as well as the pre-trained datasets. Therefore, compared with the previous setting, the white-box scenario poses a greater challenge to the defense robustness of our proposed method.

Specifically, we conduct 60 experimental settings, covering multiple self-supervised method combinations and different datasets. We use CIFAR10 as the pre-trained dataset and selected AdvEncoder as the attack method to evaluate ZePAD's defensive performance in white-box scenarios. The experimental results are provided in Tables S1 and S2.

From the experimental results, it can be seen that ZePAD maintains high RA even in white-box scenarios. For instance, when the downstream datasets are SVHN and ANIMALS10, the RA values are 89.10% and 86.80%, respectively, showing improvements of 74.66% and 45.31% over the baseline. This indicates that even with knowledge of the upstream pre-trained dataset and the pre-trained models used in the MPAD-Branch, the attacker cannot achieve effective results, demonstrating ZePAD's high robustness.

To further evaluate ZePAD's defensive performance, we tested PAP (Ban & Dong, 2022), UAP (Moosavi-Dezfooli et al., 2017), UAPGD (Deng & Karam, 2020), and SSP (Naseer et al., 2020) attacks in white-box scenarios, with results shown in Figure S1. PAP was the most stable and effective, reducing baseline RA to nearly 20%. UAP performed poorly, with baseline RA nearing 80% in some cases, such as SwAV combined with MoCo v2+. However, ZePAD maintained consistently high RA across all attacks in white-box scenarios, indicating its strong defense against existing DAEs.

Notably, we would like to emphasize that, due to the inherent uncertainty in user behavior and the vast array of available pre-trained encoders, it is extremely challenging for the attacker to accurately determine which specific two encoders are being used in practice. It is therefore extremely difficult for the attacker to perform a precise attack, which aligns with our assumed white-box scenario. As a result, the white-box setting represents the most challenging case for downstream-agnostic adversarial attacks.

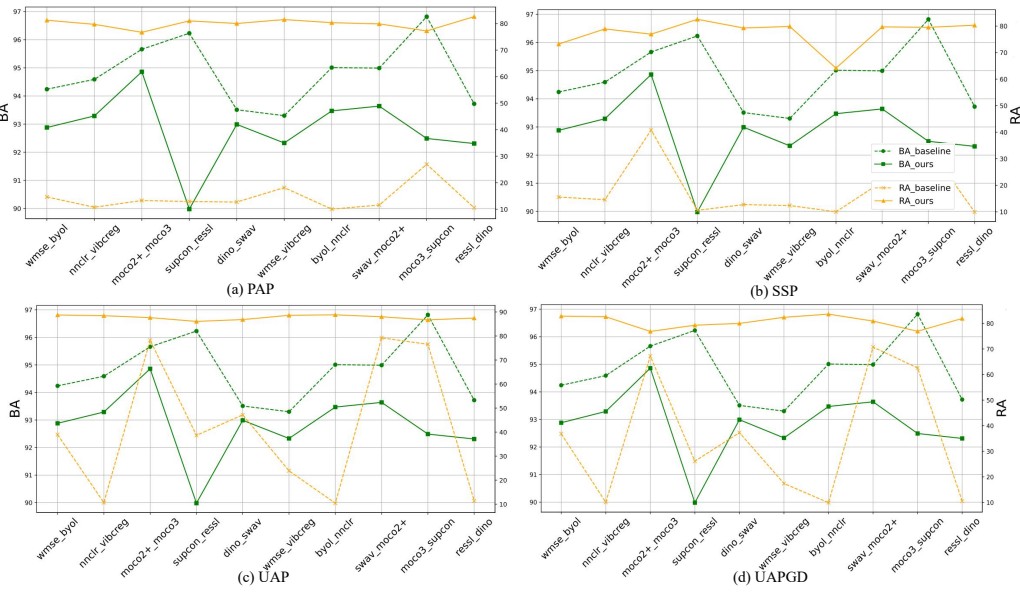

(a) PAP

(b) SSP

(c) UAP

(d) UAPGD

Figure S1: BA (%) and RA (%) of our ZePAD and the baseline method under four attack methods in the white-box scenario.

Table S1: BA(%) comparison between the baseline method and ZePAD in the white-box scenario, where $\mathcal{D}_f$ and $\mathcal{D}_d$ denote the adversarial training and downstream datasets, respectively.

| $\mathcal{D}_f$ and $\mathcal{D}_d$ | Method | W-MSE BYOL | NNCLR VibCreg | MoCo2+ MoCo3 | SupCon RESSL | DINO SwAV | W-MSE VibCreg | BYOL NNCLR | SwAV MoCo2+ | MoCo3 SupCon | RESSL DINO | AVG |
|---|---|---|---|---|---|---|---|---|---|---|---|---|
| CIFAR10 | baseline | 94.24 | 94.59 | 95.66 | 96.23 | 93.51 | 93.30 | 95.01 | 94.99 | 96.82 | 93.72 | 94.81 |
| | ours | 93.36 | 93.29 | 94.86 | 89.98 | 92.99 | 92.33 | 93.47 | 93.64 | 92.49 | 92.31 | 92.87 |
| | Δ | -0.88 | -1.30 | -0.80 | -6.25 | -0.52 | -0.97 | -1.54 | -1.35 | -4.33 | -1.41 | -1.94 |
| STL10 | baseline | 84.26 | 85.20 | 86.63 | 87.76 | 85.03 | 83.54 | 85.85 | 85.15 | 87.98 | 85.56 | 85.70 |
| | ours | 85.41 | 85.63 | 85.85 | 85.08 | 83.79 | 85.51 | 85.43 | 84.29 | 85.88 | 83.66 | 85.05 |
| | Δ | +1.15 | +0.43 | -0.78 | -2.68 | -1.24 | +1.97 | -0.42 | -0.86 | -2.10 | -1.90 | -0.64 |
| ANIMALS10 | baseline | 89.61 | 95.47 | 90.78 | 91.99 | 90.39 | 93.68 | 93.02 | 91.48 | 92.04 | 89.70 | 91.82 |
| | ours | 95.96 | 97.27 | 97.46 | 97.40 | 95.93 | 96.90 | 96.35 | 97.34 | 96.40 | 97.24 | **96.83** |
| | Δ | +6.35 | +1.80 | +6.68 | +5.41 | +5.54 | +3.22 | +3.33 | +5.86 | +4.36 | +7.54 | **+5.01** |
| GTSRB | baseline | 79.47 | 85.56 | 83.94 | 84.45 | 86.85 | 82.96 | 82.58 | 86.05 | 86.57 | 82.51 | 84.09 |
| | ours | 94.92 | 94.22 | 95.11 | 93.97 | 95.79 | 95.24 | 93.78 | 96.16 | 94.34 | 93.94 | **94.75** |
| | Δ | +15.45 | +8.66 | +11.17 | +9.52 | +8.94 | +12.28 | +11.20 | +10.11 | +7.77 | +11.43 | **+10.65** |
| ImageNet20 | baseline | 67.66 | 68.95 | 69.64 | 66.87 | 69.94 | 65.87 | 70.54 | 69.05 | 66.57 | 68.75 | 68.38 |
| | ours | 69.25 | 70.34 | 71.13 | 63.10 | 64.88 | 69.84 | 68.75 | 68.65 | 68.06 | 65.08 | 67.91 |
| | Δ | +1.59 | +1.39 | +1.49 | -3.77 | -5.06 | +3.97 | -1.79 | -0.40 | +1.49 | -3.67 | -0.48 |
| SVHN | baseline | 65.57 | 76.18 | 70.64 | 75.58 | 77.27 | 74.67 | 69.89 | 75.11 | 74.88 | 74.87 | 73.47 |
| | ours | 94.23 | 94.14 | 95.67 | 94.50 | 94.05 | 92.59 | 95.09 | 95.48 | 94.28 | 94.71 | **94.47** |
| | Δ | +28.66 | +17.96 | +25.03 | +18.92 | +16.78 | +17.92 | +25.20 | +20.37 | +19.40 | +19.84 | **+21.01** |

# G    OTHER ABLATION STUDIES

## G.1    IMPACT OF THE NUMBER OF ADVAU-MODELS

ZePAD is a multi-encoder architecture that increases diversification at the encoder level. Previous experiments have validated the effectiveness of our method. A natural idea is to introduce more multiple models to further boost performance.

To explore the impact of the number of AdvAu models on the overall model, we conducted experiments using encoders pre-trained on CIFAR10, with STL10 as the downstream dataset. We chose W-MSE as the victim encoder and considered SimCLR, BYOL, DINO, MoCo v3,

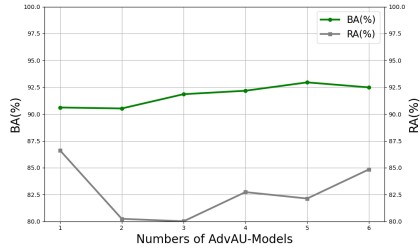

Figure S2: BA (%) and RA (%) across different numbers of AdvAu-Models.

Table S2: RA (%) comparison between the baseline method and ZePAD in the white-box scenario, where $\mathcal{D}_f$ and $\mathcal{D}_d$ denote the adversarial training and downstream datasets, respectively.

| $\mathcal{D}_f$ and $\mathcal{D}_d$ | Method | W-MSE BYOL | NNCLR VibCreg | MoCo2+ MoCo3 | SupCon RESSL | DINO SwAV | W-MSE VibCreg | BYOL NNCLR | SwAV MoCo2+ | MoCo3 SupCon | RESSL DINO | AVG |
|---|---|---|---|---|---|---|---|---|---|---|---|---|
| CIFAR10 | baseline | 14.67 | 10.77 | 13.32 | 12.93 | 12.74 | 18.17 | 10.08 | 11.60 | 26.97 | 10.58 | 14.18 |
| | ours | 81.27 | 79.73 | 76.72 | 81.03 | 80.05 | 81.52 | 80.35 | 79.90 | 77.25 | 82.63 | **80.05** |
| | Δ | +66.60 | +68.96 | +63.40 | +68.10 | +67.31 | +63.35 | +70.27 | +68.30 | +50.28 | +72.05 | **+65.86** |
| STL10 | baseline | 49.95 | 38.26 | 55.86 | 36.59 | 23.00 | 45.08 | 34.51 | 19.29 | 59.68 | 28.45 | 39.07 |
| | ours | 75.04 | 76.33 | 79.12 | 76.33 | 69.21 | 76.61 | 75.85 | 71.90 | 78.35 | 77.65 | **75.64** |
| | Δ | +25.09 | +38.07 | +23.26 | +39.74 | +46.21 | +31.53 | +41.34 | +52.61 | +18.67 | +49.20 | **+36.57** |
| ANIMALS10 | baseline | 56.06 | 50.99 | 43.53 | 54.27 | 11.37 | 52.70 | 39.97 | 13.68 | 61.46 | 30.88 | 41.49 |
| | ours | 90.88 | 87.81 | 90.18 | 91.28 | 87.97 | 88.42 | 92.14 | 66.27 | 86.05 | 87.00 | **86.80** |
| | Δ | +34.82 | +36.82 | +46.65 | +37.01 | +76.60 | +35.72 | +52.17 | +52.59 | +24.59 | +56.12 | **+45.31** |
| GTSRB | baseline | 17.56 | 10.84 | 17.27 | 16.76 | 13.85 | 16.25 | 9.49 | 9.51 | 19.27 | 5.73 | 13.65 |
| | ours | 73.45 | 75.84 | 77.27 | 70.60 | 94.08 | 76.99 | 84.69 | 93.36 | 78.05 | 82.99 | **80.73** |
| | Δ | +55.89 | +65.00 | +60.00 | +53.84 | +80.23 | +60.74 | +75.20 | +83.85 | +58.78 | +77.26 | **+67.08** |
| ImageNet20 | baseline | 37.30 | 28.08 | 24.11 | 28.08 | 8.23 | 31.85 | 21.23 | 6.15 | 40.48 | 20.73 | 24.62 |
| | ours | 61.21 | 51.79 | 61.90 | 55.46 | 39.68 | 49.70 | 62.20 | 30.75 | 53.87 | 56.94 | **52.35** |
| | Δ | +23.91 | +23.71 | +37.79 | +27.38 | +31.45 | +17.85 | +40.97 | +24.60 | +13.39 | +36.21 | **+27.73** |
| SVHN | baseline | 19.58 | 9.22 | 19.62 | 10.25 | 14.30 | 10.21 | 17.17 | 18.91 | 15.67 | 9.45 | 14.44 |
| | ours | 85.68 | 91.45 | 90.78 | 88.17 | 91.78 | 82.64 | 90.43 | 90.16 | 89.66 | 90.27 | **89.10** |
| | Δ | +66.10 | +82.23 | +71.16 | +77.92 | +77.48 | +72.43 | +73.26 | +71.25 | +73.99 | +80.82 | **+74.66** |

Table S3: Performance under different ensemble methods.

| Confidence | Metric | W-MSE | BYOL | NNCLR | SimCLR | MoCo2+ | AVG |
|---|---|---|---|---|---|---|---|
| Average | BA (%) | 93.43 | 93.61 | 93.91 | 93.95 | 93.57 | 93.69 |
| | RA (%) | 85.46 | 79.85 | 77.58 | 71.93 | 64.38 | 75.84 |
| Entropy | BA (%) | 90.97 | 90.32 | 92.07 | 91.60 | 80.36 | 89.06 |
| | RA (%) | 23.13 | 24.06 | 24.39 | 26.46 | 30.98 | 25.80 |
| RFDM | BA (%) | 93.37 | 93.49 | 93.69 | 93.70 | 93.57 | 93.56 |
| | RA (%) | 87.14 | 81.70 | 79.66 | 74.30 | 65.91 | 77.74 |

MoCo v2+, and NNCLR as potential adversarial auxiliary encoders. The results are shown in Figure S2, where we observe that increasing the number does not necessarily enhance performance. Additionally, considering the extra resource overhead introduced by incorporating multiple models, we empirically recommend setting the number of AdvAu models to one as the default choice.

## G.2 Impact of Downstream Epochs.

We study the effect of the number of epochs on the downstream model, with results shown in Figure S3. Both BA and RA remain largely unaffected by the number of epochs, which suggests that the downstream model converges quickly. This is due to the model's small size and the strong feature extraction ability of the upstream encoder. It also demonstrates that the generalization and robustness of the entire model stem primarily from the upstream model.

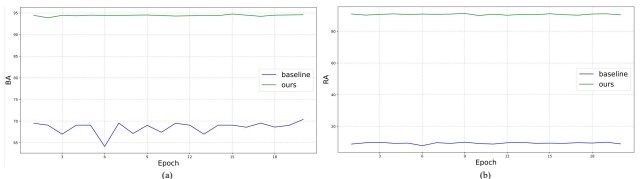

Figure S3: Performance of downstream models with different epochs. (a)-(b) denote BA (%) and RA (%) with different epochs, respectively.

## G.3 Impact of the Different Ensembles

We compare RFDM with other ensemble strategies, including average ensemble and entropy ensemble methods. In our experimental setup, the pre-training dataset for the encoder, the adversarial training dataset, and the downstream task dataset are all selected from CIFAR10. The experimental results are reported in Table S3. As shown in the results, entropy weighting performs the worst, particularly on RA. The performance of RFDM on BA is nearly identical to that of average ensemble. This is because, as illustrated in Figure 2, the confidence scores of the three branches for benign samples are almost the same, in which case RFDM is approximately equivalent to average ensemble method. In contrast, RFDM achieves slightly higher RA, which is attributed to the pronounced confidence differences between the MAPE-Branch and BME-Branch on adversarial examples. Such discrepancies allow RFDM to more effectively leverage the robustness of the MAPE-Branch when encountering adversarial examples.

Table S4: RA (%) under different perturbation budgets.

| $\epsilon$ | BYOL | ReSSL | SupCon | SimCLR | NNCLR | AVG |
|---|---|---|---|---|---|---|
| 10/255 | 80.18 | 84.58 | 77.99 | 74.70 | 79.84 | 79.46 |
| 12/255 | 74.29 | 72.41 | 71.83 | 61.80 | 70.72 | 70.21 |
| 14/255 | 57.67 | 66.41 | 68.22 | 59.32 | 67.46 | 63.82 |
| 16/255 | 53.49 | 66.53 | 61.17 | 54.45 | 53.11 | 57.75 |

Table S5: Ablation study on the effect of $\lambda$.

| $\lambda$ | W-MSE | | SupCon | | SimCLR | | NNCLR | | BYOL | |
|---|---|---|---|---|---|---|---|---|---|---|
| | BA(%) | RA(%) | BA(%) | RA(%) | BA(%) | RA(%) | BA(%) | RA(%) | BA(%) | RA(%) |
| 1 | 93.02 | 45.19 | 93.04 | 49.63 | 92.74 | 41.43 | 92.70 | 55.00 | 92.98 | 41.29 |
| 10 | 93.37 | 87.15 | 92.46 | 76.84 | 93.7 | 74.30 | 93.53 | 79.53 | 93.18 | 81.70 |
| 15 | 93.59 | 87.59 | 92.59 | 76.77 | 94.53 | 74.21 | 93.78 | 79.47 | 93.47 | 81.12 |
| 20 | 93.42 | 86.84 | 92.39 | 77.99 | 95.05 | 74.70 | 93.96 | 79.84 | 93.49 | 80.18 |
| 25 | 93.21 | 87.62 | 92.46 | 77.37 | 94.66 | 74.63 | 93.69 | 79.03 | 93.59 | 81.36 |

### G.4 IMPACT OF THE PERTURBATION BUDGETS

In the previous experiments, the perturbation budget used by the attacker was set to $10/255$ by default. To further investigate the defensive capability of ZePAD, we evaluate its performance under larger perturbation budgets. In this experimental setup, the attacker's source dataset, the encoder's pre-training dataset, the adversarial training dataset, and the downstream task dataset are all selected from CIFAR10. The experimental results are presented in Table S4. As shown in the results, RA gradually decreases as the perturbation budget increases. However, even when the perturbation budget is enlarged to 60% of the original value, RA remains above 50%, demonstrating the strong defensive performance of ZePAD.

### G.5 IMPACT OF THE PARAMETER $\lambda$

In this section, we provide an analysis of the hyperparameter $\lambda$. In the previous experiments, $\lambda$ was set to 20 by default. We investigate the performance of ZePAD as $\lambda$ varies from 1 to 25. The pre-training dataset, adversarial training dataset, and downstream dataset are all selected from CIFAR10. The experimental results are shown in Table S5. From these results, we observe that when $\lambda$ is small, the RA remains low, indicating that the encoder is insufficiently adversarially tuned. As $\lambda$ increases, the model progressively balances BA and RA, achieving a better trade-off. We therefore adopt $\lambda = 20$ as the default setting, as it provides stable performance across different settings.

## H EXTRACTED PATTERNS OF DIFFERENT SSL METHODS

As discussed previously, we find that pre-trained models based on different SSL methods may have distinct and complementary feature extraction patterns, with different SSL paradigms focusing on and memorizing different patterns. The ablation study shows that simply combining three encoders without extra adversarial training can effectively enhance both BA and RA, which supports our basic assumption that existing DAE generation methods struggle to identify common vulnerable patterns learned across different SSL methods.

To better understand these patterns, we used Grad-CAM to visualize encoders from different self-supervised methods, including BYOL,

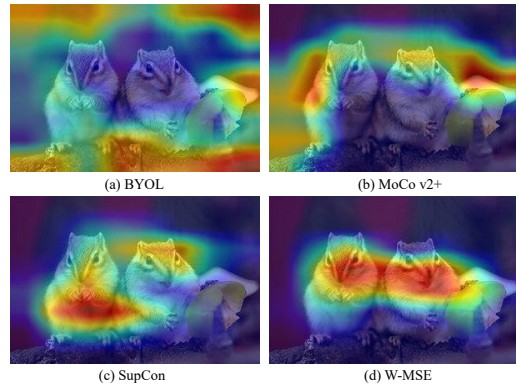

(a) BYOL      (b) MoCo v2+

(c) SupCon      (d) W-MSE

Figure S4: The Grad-CAM visualizations of different SSL methods.

MoCo v2+, SupCon, and W-MSE, all pre-trained on CIFAR10. The visualizations, shown in Figure S4, reveal that different encoders focus on different features, indicating their diverse extraction patterns and ability to handle varied samples. This diversity allows RFDM to leverage their strengths, thereby improving overall model performance.

## I  RESOURCE OCCUPANCY AND CONSUMPTION

In our proposed method, additional encoders are incorporated. It is expected that this would lead to increased resource consumption and occupancy, including inference time and GPU memory. Given that our method only utilizes multiple encoders during the inference phase, we conducted evaluations specifically in this stage. The experiments were conducted on CIFAR10 using an NVIDIA RTX 4090 GPU. We performed inference on the test sets and measured the average time overhead and GPU memory usage per batch, with a batch size of 256.

The results are presented in Table S6. Our method introduces additional computational overhead; however, the extra inference time is only 0.00007 seconds per image, which represents approximately a 30% increase compared to the

Table S6: The Computational Resource Comparison.

| Method | Time (s) | Memory (MB) |
|---|---|---|
| Baseline | 0.0542 | 67.04 |
| ZePAD | 0.0714 | 156.50 |

baselines. We believe this overhead is acceptable given the substantial performance improvements. Considering the substantial improvements in both BA and RA (over 30% and 80% respectively in some cases), we believe that the additional computational cost is justified and acceptable.

While our method introduces multiple branches and therefore incurs a higher training cost per run, it only requires a single adversarial training process for different downstream tasks. In contrast, existing methods typically require separate adversarial training for each downstream dataset. Hence, in practical scenarios involving multiple downstream tasks, our approach can be more efficient overall. In particular, when the number of downstream tasks exceeds the number of branches, the total training cost of our method becomes lower than that of task-specific adversarial training.

In our future work, we will focus on reducing the computational overhead while further preserving both robustness and benign accuracy.

## J  EXTENSION TO OTHER DOWNSTREAM TASKS

To further evaluate the performance of ZePAD on other downstream tasks, we design a retrieval experiment. The encoder is pre-trained on CIFAR10, while both the adversarial training dataset and the retrieval dataset are chosen from SVHN. In this experiment, the training set is used as the gallery and the test set is used as the query set. Table S7 reports the Top-10 accuracy of this task. From the results, we observe that ZePAD achieves substantial improvements in both BA and RA compared to baseline, indicating that ZePAD retains its zero-sacrifice property on alternative downstream tasks.

Table S7: ZePAD's Performance on the Retrival Task.

| Metric | method | W-MSE | BYOL | NNCLR | SimCLR | MoCo2+ | MoCo3 | ReSSL | DINO | SwAV | VibCreg | SupCon | AVG |
|---|---|---|---|---|---|---|---|---|---|---|---|---|---|
| BA (%) | baseline | 62.25 | 62.19 | 58.85 | 60.10 | 69.05 | 70.22 | 62.89 | 56.32 | 69.73 | 75.47 | 59.74 | 64.26 |
|  | ZePAD | 90.49 | 86.74 | 84.53 | 88.79 | 89.00 | 90.20 | 78.97 | 80.37 | 81.00 | 95.26 | 88.85 | 86.75 |
| RA (%) | baseline | 50.64 | 58.41 | 51.25 | 39.37 | 67.42 | 64.80 | 57.38 | 67.17 | 61.12 | 64.70 | 53.13 | 57.76 |
|  | ZePAD | 75.13 | 80.93 | 77.03 | 80.34 | 74.71 | 73.91 | 68.71 | 71.89 | 81.82 | 87.92 | 85.78 | 78.02 |

While extending it to tasks such as detection or segmentation is theoretically possible, these tasks follow fundamentally different architectural pipelines (e.g., U-Net variants) and heavily rely on skip connections or task-specific modules. Directly replacing their encoders with a standalone pre-trained encoder would likely lead to suboptimal performance, and thus this direction is beyond the current scope of this paper. We appreciate this insightful suggestion and intend to explore the extension of our method to segmentation models, such as SAM, as part of future research.

Table S8: BA (%) comparison of Baseline and ZePAD with CLIP pre-trained encoder.

| Dataset | Method | W-MSE | SimCLR | NNCLR | MoCo2+ | MoCo3 | AVG |
|---------|--------|-------|--------|-------|--------|-------|-----|
| CIFAR10 | Baseline | 90.17 | 93.56 | 93.77 | 95.03 | 94.71 | 93.45 |
| | ZePAD | 93.95 | 92.63 | 94.31 | 95.22 | 94.23 | 94.07 |
| STL10 | Baseline | 78.49 | 81.94 | 83.09 | 84.46 | 84.00 | 82.40 |
| | ZePAD | 95.07 | 96.64 | 97.33 | 97.36 | 97.68 | 96.82 |
| ANIMALS10 | Baseline | 80.39 | 90.82 | 91.22 | 88.03 | 88.05 | 87.70 |
| | ZePAD | 99.33 | 98.24 | 99.17 | 98.89 | 99.45 | 99.02 |

Table S9: RA (%) comparison of Baseline and ZePAD with CLIP pre-trained encoder.

| Dataset | Method | W-MSE | SimCLR | NNCLR | MoCo2+ | MoCo3 | AVG |
|---------|--------|-------|--------|-------|--------|-------|-----|
| CIFAR10 | Baseline | 10.09 | 47.13 | 10.71 | 45.01 | 36.86 | 29.96 |
| | ZePAD | 79.99 | 81.24 | 82.35 | 79.34 | 78.66 | 80.32 |
| STL10 | Baseline | 41.16 | 67.52 | 44.05 | 65.55 | 64.28 | 56.51 |
| | ZePAD | 92.65 | 91.73 | 92.78 | 91.04 | 92.33 | 92.11 |
| ANIMALS10 | Baseline | 48.31 | 63.48 | 49.23 | 63.03 | 60.32 | 56.87 |
| | ZePAD | 97.26 | 96.63 | 97.28 | 95.34 | 97.33 | 96.77 |

## K  EXPERIMENTS ON MULTIMODAL PRE-TRAINED ENCODERS

To investigate the performance of ZePAD on multimodal and larger models, we conduct experiments based on CLIP. We use CLIP as the MAPE-Branch to protect different pre-trained encoders, including W-MSE, SimCLR, NNCLR, MoCo v2+, and MoCo v3. Both CLIP variants, ResNet-101-based and ViT-B/32-based, are adversarial fine-tuned on CIFAR10 with a batch size of 64, and the hyperparameter $\lambda$ and learning rate are set to 20 and 0.00001, respectively. The experimental results are presented in Tables S8 and S9, respectively.

Even with CLIP, a large-scale multimodal pre-trained encoder, our method preserves benign accuracy while achieving a substantial improvement in adversarial robustness. These results demonstrate the generalizability of our approach to multimodal pre-trained encoders. The main reason is that the fundamental threat model of downstream-agnostic adversarial attacks remains consistent across both single-modality and multimodality encoders, and the inherent sensitivity leveraged by our defense strategy does not depend on the modality used in pre-training but rather on the distribution alignment between samples and the encoder's training data.

