# OpenReview forum: "Zero-Sacrifice Persistent-Robustness Adversarial Defense for Pre-Trained Encoders"
_ICLR.cc/2026/Conference — ICLR 2026 Poster_

### Official Review · Reviewer_pqbZ · 2025-10-28

**Soundness:** 3
**Presentation:** 2
**Contribution:** 2
**Rating:** 4
**Confidence:** 4

**Summary:**

This paper proposes ZeLAD, a “Zero-Sacrifice Lifelong Adversarial Defense” framework that integrates multiple adversarial and benign branches to achieve both clean accuracy and robustness without further tuning. The method designs a Robust Federal Decision Mechanism (RFDM) to adaptively weight branch predictions based on confidence, aiming to provide a single-tuning defense applicable across tasks. Experiments are conducted on multiple datasets and architectures.

**Strengths:**

1. The experiments are relatively comprehensive, covering several datasets, model scales, and attack types.
2. The paper is logically organized, and most sections follow a clear structure.
3. The proposed method is systematic and addresses the problem of balancing robustness and generalization in a targeted manner.

**Weaknesses:**

1. Core claim overstatement: The claim of “lifelong + zero-sacrifice with single tuning” is not well supported. Although the paper emphasizes a one-time tuning process, in practice each new downstream task still requires training local classifiers and possibly branch adaptation. The experiments only test cross-dataset transfer, not true lifelong learning without any retraining, making the central claim over-stated.
2. Over-packaged novelty: The technical novelty is limited. ZeLAD essentially combines multiple encoder branches with a handcrafted confidence-weighted ensemble (RFDM). The exponential weighting is heuristic, and there is no comparison with simpler ensemble or calibration baselines. The contribution is more about system integration than a fundamentally new defense principle.
3. Experimental and analysis issues:
    - It is suggested to include a comparison with a three-encoder average ensemble or PGD-AT baseline to validate RFDM’s effectiveness.
    - Hyperparameter choices (e.g., weighting coefficients, confidence scaling) and sensitivity analysis are not explained.
4. Writing and presentation problems:
    - Figure 2 should be moved to Section 3.2.2.
    - Eqs. (5) and (6) are unclear in mathematical form and explanation.
    - Unify terminology: use adversarial example consistently.
    - The introduction mentions RFDM, but later sections use inconsistent terms.
    - The PGD-AT citation is incorrect; it should be *“Towards Deep Learning Models Resistant to Adversarial Attacks”*, ICLR 2018, not the CW attack paper.
5. Mathematical writing issues:
    - Use calligraphic letters (e.g., $\mathcal{D}$) for datasets, not $D$; and use $D$ for distance.
    - Keep notations of encoder and classifier consistent ($E,F$ in Section 3.1 while $\mathcal{E},\mathcal{F}$ in Section 3.2).
    - Avoid blank lines after \begin{equation}.
    - Rewrite Eqs. (5) and (6) with explicit meaning of each symbol.
    - Annotate dimensionality for key notations if possible.
    - For clarity, it is suggested to include corresponding notation for each component in Figure 1.
6. Typos and minor language errors:
    - “generalibility” → “generalizability” (p.1 l.16)
    - “differ with” → “different from” (p.1 l.18)
    - “... uses two” → “that uses two” (p.1 l.23)
    - “task-specfic” → “task-specific” (p.2 l.75)
    - “taht” → “that”; “branche” → “branches” (p.9 l.446)
    - Algorithm 1 needs a thorough proofreading.
7. Related work limitations: The related work section is short and somewhat outdated. Please include more recent adversarial defense papers, such as:
    - *Probabilistic Margins for Instance Reweighting in Adversarial Training*, NeurIPS 2021.
    - *DAT: Improving Adversarial Robustness via Generative Amplitude Mix-up in Frequency Domain*, NeurIPS 2024

**Questions:**

See Weaknesses. If the authors address my concerns, I am willing to raise the score.

---

> ### Author Response · Authors · 2025-11-23
> **Response to Reviewer pqbz(1/2)**
>
> We sincerely appreciate the reviewer’s careful and constructive comments. Below, we provide detailed responses to each concern.
>
> > **W1:** Statement about "Lifelong + Zero-Sacrifice": The claim of “lifelong + zero-sacrifice with single tuning” is not well supported.
>
>
> **Response:** Thank you for your insightful question. Our method leverages an inherent property of neural networks: they tend to assign higher confidence to samples that align with their training data distribution. Building on this, our proposed RFDM mechanism further amplifies the confidence discrepancy between the two branches, ensuring that for each input, the branch trained on the most relevant distribution naturally dominates the final decision.
>
> This design yields two key benefits:
>
> (1) It guarantees a performance lower bound, since benign and adversarial examples are respectively handled by the branch that is most sensitive to their distribution;
>
> (2) When both branches provide comparable predictions (e.g., under stronger or adaptive attacks), the fusion mechanism produces a more robust and balanced response rather than performance degradation.
>
> These benefits are the reason why our method can achieve the "lifelong" property.
>
> Besides, we understand that the term "lifelong" may lead to potential misunderstanding with continual learning. In our paper, "lifelong" does not refer to incremental task adaptation, but instead to the ability of a single adversarially trained model to be applied across different downstream tasks without retraining or task-specific fine-tuning.
>
> In contrast to existing downstream-agnostic adversarial defense methods, which require separate adversarial training for each downstream task, our method only performs adversarial training once on a single dataset, and then our method can be directly transferred to other unseen downstream tasks while maintaining robustness. This "train-once, defend-forever" property is what we refer to as lifelong in our context.
>
> To avoid confusion, we have revised the term to “Persistent-Robustness” to avoid unnecessary misunderstandings.
>
> ---
>
> > **W2:** The technical novelty is limited. The contribution is more about system integration than a fundamentally new defense principle.
>
>
>
> **Response:**  Thank you for your comment. The core novelty lies in revealing and theoretically formalizing an intrinsic property of neural networks: their inherent sensitivity to data distributions, which manifests as confidence separation across differently trained branches. Based on this insight, we proposed RFDM, which is not a heuristic ensemble, but a mechanism that amplifies distribution-induced confidence discrepancies to allow the most distribution-aligned branch to dominate the decision. Our method is simple but effective, and the architectural details serve only as a practical instantiation to realize our core idea.
>
> More importantly, our work introduces a new threat model that requires not only defending against adversarial examples but also preserving benign accuracy simultaneously, which goes beyond traditional robustness-only objectives.
>
> ---
>
>
> > **W3:** Experimental and analysis issues: 1. Insufficient ablation study on RFDM; 2. Lack of experiments regarding hyperparameter selection.
>
>
> **Response:** Thank you for your suggestion. To empirically validate the effectiveness of the hyper-parameter choice, we have conducted a series of experiments in the revised manuscript. Specifically, we compare RFDM with other ensemble strategies, including average ensemble and entropy ensemble methods, and the results are shown below. It can be noticed that RFDM can achieve the best RA in all settings, which illustrates the effectiveness of our proposed method. Besides, the performance of RFDM on BA is nearly identical to that of average ensemble. This is because, as illustrated in Figure 2 of our main text, the confidence scores of the three branches for benign samples are almost the same, in which case RFDM is approximately equivalent to the average ensemble method. The detailed discussions can be found in **Appendix G.3**.
>
>
>
> **Table 1. Performance under different ensemble methods.**
>
> | Confidence | Metric | W-MSE | BYOL  | NNCLR | SimCLR | MoCo2+ | AVG   |
> | ---------- | ------ | ----- | ----- | ----- | ------ | ------ | ----- |
> | Average    | BA (%) | 93.43 | 93.61 | 93.91 | 93.95  | 93.57  | 93.69 |
> |            | RA (%) | 85.46 | 79.85 | 77.58 | 71.93  | 64.38  | 75.84 |
> | Entropy    | BA (%) | 90.97 | 90.32 | 92.07 | 91.60  | 80.36  | 89.06 |
> |            | RA (%) | 23.13 | 24.06 | 24.39 | 26.46  | 30.98  | 25.80 |
> | RFDM       | BA (%) | 93.37 | 93.49 | 93.69 | 93.70  | 93.57  | 93.56 |
> |            | RA (%) | 87.14 | 81.70 | 79.66 | 74.30  | 65.91  | 77.74 |

---

> ### Author Response · Authors · 2025-11-23
> **Response to Reviewer pqbz(2/2)**
>
> Furthermore, we provide a sensitivity analysis of the hyper-parameter $\lambda$ in **Appendix G.5**, and the corresponding results are shown below. From these results, we observe that when $\lambda$ is small, the RA remains low, indicating that the encoder is insufficiently adversarially tuned. As $\lambda$ increases, the model progressively balances BA and RA, achieving a better trade-off. We therefore adopt $\lambda = 20$ as the default setting, as it provides stable performance across different settings. The detailed discussions can be found in **Appendix G.5**. To help readers reproduce our method, we give all hyper-parameters in **Appendix E**, and we will release our code after the review phase.
>
>
>
> **Table 2. Ablation study on the effect of $\lambda$.**
>
> | $\lambda$ | W-MSE BA(%) | W-MSE RA(%) | SupCon BA(%) | SupCon RA(%) | SimCLR BA(%) | SimCLR RA(%) | NNCLR BA(%) | NNCLR RA(%) | BYOL BA(%) | BYOL RA(%) |
> | --------- | ----------- | ----------- | ------------ | ------------ | ------------ | ------------ | ----------- | ----------- | ---------- | ---------- |
> | 1         | 93.02       | 45.19       | 93.04        | 49.63        | 92.74        | 41.43        | 92.70       | 55.00       | 92.98      | 41.29      |
> | 10        | 93.37       | 87.15       | 92.46        | 76.84        | 93.70        | 74.30        | 93.53       | 79.53       | 93.18      | 81.70      |
> | 15        | 93.59       | 87.59       | 92.59        | 76.77        | 94.53        | 74.21        | 93.78       | 79.47       | 93.47      | 81.12      |
> | 20        | 93.42       | 86.84       | 92.39        | 77.99        | 95.05        | 74.70        | 93.96       | 79.84       | 93.49      | 80.18      |
> | 25        | 93.21       | 87.62       | 92.46        | 77.37        | 94.66        | 74.63        | 93.69       | 79.03       | 93.59      | 81.36      |
>
>
> ---
>
> > **W4, W5, W6:** Writing and presentation problems, Mathematical writing issues, and Typos and minor language errors.
>
> **Response**
> Thank you for your comment. We have moved Fig. 2 to Sec. 3.2.2 and rewritten Eqs. (5) and (6) with full explanations. In addition, we have unified the notations, corrected the citations, and revised the mathematical expressions according to your suggestion. We have also added missing notations in Fig. 1. Finally, the manuscript has been thoroughly proofread to correct typographical and formatting errors.
>
>
> ---
>
> > **W7:** Related work limitations.
>
> **Response**
> Thanks for your comment. Due to page limitations in the main text, we provided only a high-level discussion of related work to maintain focus and ensure the core method remains clear to readers. The detailed related works are included in the revised appendix. Additionally, in the revised version, we have incorporated relevant references such as "Probabilistic Margins for Instance Reweighting in Adversarial Training" and "DAT: Improving Adversarial Robustness via Generative Amplitude Mix-up in Frequency Domain" to further strengthen the discussion.

---

> ### Comment · Reviewer_pqbZ · 2025-11-26
> **Minor Issues**
>
> Thank you for the revised version. Most of my earlier concerns have been properly addressed, and the current manuscript is significantly improved. I only have some minor comments that may further enhance the clarity and presentation quality:
> 1. For clarity and in accordance with ICLR’s guideline, please highlight all modifications in the revision (e.g., using blue color).
>
> 2. ICLR encourages including an *Ethics Statement* and a *Reproducibility Statement* at the end of the main paper. I recommend adding both sections.
>
> 3. The layout still appears unusual, with several places where the spacing is too tight due to frequent manual `\vspace` adjustments, e.g.,
>    * between Fig. 1 and Sec. 3.3
>    * between Table 1 and the following text
>    * between Table 2 and Sec. 4
>    * above Table 3 caption, and between Table 3 and the text below
>    * above Table 6 caption, and between Table 6 and the following text
>    * spacing between all table bodies and their captions
>
>    I suggest (i) reducing paragraphs that end with only a few words on the last line, and (ii) avoiding blank lines before LaTeX `\begin{equation}` environments (e.g., Eqs. (3)–(6)) to save space and stabilize line breaking.
>
> 4. Consider splitting Sec. 3.1–3.3 into a standalone section, and keep Sec. 3.4 as a separate “Methodology” section for better structural clarity.
>
> 5. In Sec. 3.1, `E` vs. `\mathcal{E}` and `F` vs. `\mathcal{F}` are still mixed. I recommend consistently using the calligraphic forms. Also, it would be clearer to use $\phi$ as the parameter of $\mathcal{F}$ (to differentiate it from $\mathcal{E}$'s parameter $\theta$), and include explicit parameter subscripts in Eq. (4).
>
> 6. Typos:
>    * p. 5, line 237: $cos$ → $\mathrm{cos}$
>    * Fig. 3: ZeLAD → ZePAD
>    * Appx. C: ZeLDA → ZePAD
>    * Tables 1–3: semi–black-box → semi-black-box
>    * Algorithm 1, line 5: `\mathcal{D}_{adv} \bigcup x^*` → `\mathcal{D}_{adv} \cup \{x^*\}`
>    * Unify pretrained vs. pre-trained and pretraining vs. pre-training
>    * Unify CIFAR10 vs. CIFAR-10
>
> 7. In the bibliography, *“Pre-trained adversarial perturbations”* and *“Downstream-agnostic adversarial examples”* appear more than once.

---

> ### Author Response · Authors · 2025-11-26
> **Response to Reviewer pqbZ**
>
> Thanks for the response and we are happy that our previous responses have been properly addreseed most of your concerns.
> We sincerely appreciate the reviewer’s very careful and detailed reading of our manuscript. The level of attention and the mentioned details were surprising, and we are truly grateful. We apologize for the typos and have fixed all of them in the revised version.
>
> Thank you again for your suggestion, and our responses to each of the suggestions are provided below:
>
> > For clarity and in accordance with ICLR’s guideline, please highlight all modifications in the revision (e.g., using blue color).
>
> Thank you for the suggestion. We have highlighted all modified contents in blue.
>
> > ICLR encourages including an *Ethics Statement* and a *Reproducibility Statement* at the end of the main paper. I recommend adding both sections.
>
> Thank you for the suggestion. We have added both the *Ethics Statement* and the *Reproducibility Statement* to the main paper. They are included as Sections 7 and 8 in the revised manuscript.
>
> > The layout still appears unusual, with several places where the spacing is too tight due to frequent manual adjustments, e.g.,`\vspace`
>
> Thank you for the suggestion. Following your advice, we have adjusted the layout issues and eliminated the frequent manual formatting adjustments.
>
> > Consider splitting Sec. 3.1–3.3 into a standalone section, and keep Sec. 3.4 as a separate “Methodology” section for better structural clarity.
>
> Thank you for the suggestion. We have reorganized the section structure accordingly to improve clarity and readability.
>
> > In Sec. 3.1, vs. and vs. are still mixed. I recommend consistently using the calligraphic forms. Also, it would be clearer to use `E``\mathcal{E}``F``\mathcal{F}` as the parameter of (to differentiate it from 's parameter ), and include explicit parameter subscripts in Eq. (4).
>
> Thank you for the suggestion. We have carefully reviewed all formulas and ensured consistency throughout.
>
> > Typos
>
> Thank you for pointing these out. We have thoroughly proofread the manuscript and corrected all the typos.
>
> > In the bibliography, *“Pre-trained adversarial perturbations”* and *“Downstream-agnostic adversarial examples”* appear more than once.
>
> Thank you for your comment. We have revised the bibliography to ensure that no duplicated entries remain.

---

> > ### Comment · Reviewer_pqbZ · 2025-11-26
> >
> > Thank you for your active revisions. The readability and clarity of the manuscript have been greatly improved compared to the original version, and I believe the current version now meets the ICLR acceptance bar. Accordingly, I will raise my score to 6.
> >
> > I only want to mention two minor remaining issues that were not addressed:
> > - Unify “semi–black-box” in the captions of Tables 1, 2, and 3 to “semi-black-box” (replace the em-dash `--` with the en-dash `-` in LaTeX).
> > - In Algorithm 1, line 5, `x^*` should be replaced with **`\{x^*\}`** in LaTeX so that the union is taken with a set rather than a single element.
> >
> > Other than these tiny points, I have no further concerns. Good luck with your submission!

---

> > > ### Author Response · Authors · 2025-11-27
> > >
> > > Thank you for your careful and detailed suggestions. We have addressed them in the revision, and we truly appreciate the time and effort you invested in reviewing our work.

---

### Official Review · Reviewer_gcvV · 2025-10-29

**Soundness:** 2
**Presentation:** 3
**Contribution:** 3
**Rating:** 6
**Confidence:** 4

**Summary:**

This paper addresses the vulnerability of self-supervised learning (SSL) models to downstream-agnostic adversarial examples (DAEs). The authors propose Zero-Sacrifice Lifelong Adversarial Defense (ZeLAD), a dual-branch framework designed to achieve adversarial robustness without sacrificing benign performance. ZeLAD integrates a Multi-Pattern Adversarial Enhancement branch and a Benign Memory Preservation branch to balance robustness and benign performance. It detects DAEs by evaluating branch confidence, eliminating the need for adversarial sample training. Extensive experiments on 11 SSL methods and 6 datasets show substantial improvements in both benign accuracy and adversarial resistance, validating its “zero-sacrifice” property.

**Strengths:**

1. The paper introduces ZeLAD, the first lifelong adversarial defense for pre-trained encoders that achieves robustness across multiple downstream tasks with a single tuning. Unlike prior task-specific adversarial training methods, ZeLAD generalizes effectively across SSL models and datasets, marking a substantial conceptual advancement in adversarial robustness research.
2. A major strength is ZeLAD’s dual-branch architecture: the Multi-Pattern Adversarial Enhancement (MPAE) branch for robustness and the Benign Memory Preservation (BMP) branch for maintaining clean-sample accuracy. This design enables the model to enhance adversarial defense without degrading benign performance, a key limitation of previous methods.

**Weaknesses:**

1. Although the paper compares ZeLAD to several classic defenses (e.g., TRADES, MART, Gen-AF), it does not include enough comparisons with the most recent or SSL-specific adversarial defense methods (only Table 7). This omission makes it harder to gauge ZeLAD’s relative progress within the latest research landscape.
2. The proposed approach requires multiple encoders and dual-branch inference, which could increase computational and memory overhead compared to single-encoder defenses. The paper provides limited discussion or quantitative evaluation of training/inference efficiency. I think it is very important for real world deployment.
3. Although the paper claims “lifelong” robustness, the experiments only cover a limited number of sequential tasks. There is no long-term continual learning evaluation (e.g., over dozens of tasks or domain shifts), so the claim of lifelong adaptation remains somewhat speculative.

**Questions:**

1. The paper introduces a weighting parameter $\lambda$ in the loss function (Eq. 3), but its value or tuning procedure is not specified.
2. Some hyperparameters (e.g., lambda, learning rate) are not explicitly stated.
   Will the authors release code or configuration files to ensure experimental reproducibility?
3. The paper uses a dual-branch architecture with independent encoders. Have the authors considered some techniques to reduce redundancy while preserving robustness?

---

> ### Author Response · Authors · 2025-11-23
> **Response to Reviewer gcvV(1/2)**
>
> We sincerely appreciate the reviewer’s careful and constructive comments. Below, we provide detailed responses to each concern.
>
> > **W1:** Limited comparison with recent SSL-specific defenses.
> > Although the paper compares ZeLAD to several classic defenses (e.g., TRADES, MART, Gen-AF), it does not include enough comparisons with the most recent or SSL-specific adversarial defense methods (only Table 7). This omission makes it harder to gauge ZeLAD’s relative progress within the latest research landscape.
>
> **Response:** Thank you for your valuable suggestion. While downstream-agnostic adversarial attacks pose significant challenges to pre-trained encoder-based models, existing defense methods explicitly designed for this setting are still very limited. Among them, Gen-AF is, to the best of our knowledge, the most representative and strongest baseline, as illustrated in Table 7 of our main text. We acknowledge that there may be other relevant defense methods, and we would be happy to include additional comparisons if the reviewer could point out any important methods we might have missed.
>
>
> Besides, the comparison verifies the effectiveness of our method, where our approach consistently outperforms existing defense methods. Notably, both BA and RA are approximately 10\% higher than the second-best method across different settings. Moreover, unlike other methods that require task-specific adversarial tuning for each downstream task, our method only needs a one-time adversarial tuning. This highlights one of our key contributions: we introduce a novel threat model and demonstrate that our defense framework can achieve robust performance.
>
>
>
> ---
>
> >**W2:** Computational Cost and Practicality
> >The proposed approach requires multiple encoders and dual-branch inference, which could increase computational and memory overhead compared to single-encoder defenses. The paper provides limited discussion or quantitative evaluation of training/inference efficiency. I think it is very important for real world deployment.
>
> **Response:** Thank you for your comment. Our method increases the average inference time per image by only 0.00007 seconds. We admit that this represents a 30\% increase compared to the baselines, but the additional latency remains acceptable. Considering the substantial improvements in both BA and RA (over 30\% and 80\% respectively in some cases), we believe that this additional computational cost is justified and acceptable. In our future work, we will focus on reducing the computational overhead while further preserving both robustness and benign accuracy.
>
> While our method introduces multiple branches and therefore incurs a higher training cost per run, it only requires a single adversarial training process for different downstream tasks. In contrast, existing methods typically require separate adversarial training for each downstream dataset. Hence, in practical scenarios involving multiple downstream tasks, our approach can be more efficient overall. In particular, when the number of downstream tasks exceeds the number of branches, the total training cost of our method becomes lower than that of task-specific adversarial training.
>
> ---
>
>
> > **W3:** "Lifelong" Statement.
>
> **Response:** Thank you for your comment. We understand that the term “lifelong” may lead to potential misunderstanding with the concept of continual learning. In our paper, “lifelong” does not refer to incremental task adaptation, but rather than to the ability of a single adversarially trained model to operate across different downstream tasks without retraining or task-specific fine-tuning.
>
> In contrast to existing downstream-agnostic adversarial defense methods, which require separate adversarial training for each downstream task, our method performs adversarial training only once on a single dataset, after which it can be directly transferred to unseen downstream tasks while maintaining robustness. This “train-once, defend-forever” property is what we refer to as lifelong in our context.
>
> To avoid confusion, we have revised the term to “Persistent-Robustness” to avoid unnecessary misunderstandings.

---

> ### Author Response · Authors · 2025-11-23
> **Response to Reviewer gcvV(2/2)**
>
> > **Q1:** The ablation study on the impact of the hyperparameter $\lambda$.
>
> **Response:** Thank you for your comment. In our experiments, the default value of $\lambda$ is set to 20. Additionally, we provide a sensitivity analysis of the hyperparameter $\lambda$ in **Appendix G.5**, and the corresponding results are shown below. From these results, we observe that when $\lambda$ is small, the RA remains low, indicating that the encoder is insufficiently adversarially tuned. As $\lambda$ increases, the model progressively balances BA and RA, achieving a better trade-off. We therefore adopt $\lambda = 20$ as the default setting, as it provides stable performance across different settings.
>
> We have carefully proofread the manuscript and give all hyperparameters in **Appendix E**. Besides, the related codes will be released after the double-blind review to promote reproducibility and facilitate a clearer understanding of our method.
>
>
>
> **Table 1. Ablation study on the effect of $\lambda$.**
>
> | $\lambda$ | W-MSE BA(%) | W-MSE RA(%) | SupCon BA(%) | SupCon RA(%) | SimCLR BA(%) | SimCLR RA(%) | NNCLR BA(%) | NNCLR RA(%) | BYOL BA(%) | BYOL RA(%) |
> | --------- | ----------- | ----------- | ------------ | ------------ | ------------ | ------------ | ----------- | ----------- | ---------- | ---------- |
> | 1         | 93.02       | 45.19       | 93.04        | 49.63        | 92.74        | 41.43        | 92.70       | 55.00       | 92.98      | 41.29      |
> | 10        | 93.37       | 87.15       | 92.46        | 76.84        | 93.70        | 74.30        | 93.53       | 79.53       | 93.18      | 81.70      |
> | 15        | 93.59       | 87.59       | 92.59        | 76.77        | 94.53        | 74.21        | 93.78       | 79.47       | 93.47      | 81.12      |
> | 20        | 93.42       | 86.84       | 92.39        | 77.99        | 95.05        | 74.70        | 93.96       | 79.84       | 93.49      | 80.18      |
> | 25        | 93.21       | 87.62       | 92.46        | 77.37        | 94.66        | 74.63        | 93.69       | 79.03       | 93.59      | 81.36      |
>
>
>
> ---
>
> > **Q2:** Some hyperparameters are not explicitly stated. Will the authors release code or configuration files to ensure experimental reproducibility?
>
> **Response:** Thank you for your question. We have clarified all hyperparameters in the revised appendix. Besides, we will release all codes to ensure reproducibility after the double-blind review.
>
> ---
>
> > **Q3:** The paper uses a dual-branch architecture with independent encoders. Have the authors considered some techniques to reduce redundancy while preserving robustness?
>
> **Response:** Thank you for the helpful suggestion. In this work, our primary focus is to introduce a new defense perspective and demonstrate the effectiveness of our proposed framework under the downstream-agnostic encoder attack setting. Once the concept and its defensive capability are validated, we agree that improving computational efficiency becomes an important next step. In future work, we plan to explore techniques such as model pruning, knowledge distillation, and encoder compression to reduce redundancy while maintaining robustness. We believe these directions are promising for further enhancing the practicality of our method.

---

> > ### Comment · Reviewer_gcvV · 2025-11-24
> >
> > The rebuttal sufficiently addressed my comments. I don't have other questions.

---

> > > ### Author Response · Authors · 2025-11-24
> > > **Response to Reviewer gcvV**
> > >
> > > Thank you for your kind response. We’re very glad that our revisions have satisfactorily addressed your comments. If you have any further concerns or questions regarding our work, please feel free to let us know. We will do our best to respond during the author–reviewer discussion period.

---

### Official Review · Reviewer_2cmk · 2025-10-31

**Soundness:** 2
**Presentation:** 2
**Contribution:** 3
**Rating:** 4
**Confidence:** 5

**Summary:**

This paper introduces ZeLAD, a lifelong defense framework that protects self-supervised encoders from downstream-agnostic adversarial examples (DAEs) with a single tuning. ZeLAD employs dual branches to enhance adversarial robustness while preserving benign performance, and can even detect DAEs via branch confidence without explicit training.

**Strengths:**

1、The paper proposes a zero-sacrifice, lifelong adversarial defense method that not only maintainsbut also improves benign performance, while enhancing adversarial robustness.

2、Extensive experimental results demonstrate the effectiveness of the proposed method.

3、The paper is easy to follow.

**Weaknesses:**

1、The paper claims to build on the inherent sensitivity of neural networks to data characteristics, yet this idea is only briefly mentioned in the introduction (L54–57) without deeper investigation. No experimental validation, theoretical analysis, or concrete insight is provided to support this claim, which substantially undermines the rationale and validity of the proposed method. A more thorough analysis through exploratory experiments or theoretical justification is necessary.

2、The first and second claimed contributions both emphasize lifelong adversarial defense, which appear conceptually identical. I suggest merging them for conciseness and clarity.

3、The paper inconsistently uses adversarial sample and adversarial example. The terminology should be unified using the widely accepted adversarial example to maintain professional consistency.

4、The Related Work section remains too high-level and lacks discussion of key concepts (e.g., “Pre-trained encoders”， “pre-trained paradigm”) as well as core algorithmic ideas of representative methods—such as self-supervised learning approaches (SimCLR, MoCo) and DAEs on pre-trained encoders (PAP, AdvEncoder).

5、The Threat Model is essential for any security-related study and should appear in the main text rather than the appendix. Similarly, the discussion of challenges addressed by the proposed method would be better placed before the methodology section for improved logical flow.

6、The reported RA results (Tables 2 and 4) show large discrepancies compared to baseline values, raising concerns about correct reproduction of baseline methods. The causes of these gaps should be clarified.

7、The proposed method demonstrates higher time and storage costs than the baselines (Table S3, L899–916).

8、The overall writing quality requires improvement, particularly in table captions, many of which contain grammatical errors. For instance, “Table 1: BA(%) Baseline vs. ZeLAD in the semi-black-box scenario” is ungrammatical and should be revised for correctness and clarity.

**Questions:**

Given that the proposed method is claimed to build on a general property of deep neural networks — “neural networks inherently exhibit higher confidence in inputs that resemble the training data, a behavior attributed to the memorization of the data’s characteristics” — two questions arise.

1. Whether the method can be applied to multimodal pre-trained models such as CLIP or BLIP remains unclear. Experiments are needed to verify its scalability.

2. It is also unclear whether the method can generalize beyond image classification, for example to image retrieval, semantic segmentation, or object detection.

---

> ### Author Response · Authors · 2025-11-23
> **Response to Reviewer 2cmk(1/4)**
>
> We sincerely appreciate the reviewer’s careful and constructive comments. Below, we provide detailed responses to each concern.
>
> > **W1:** Theoretical analysis about the proposed method.
>
>
> **Response:** Thank you for your suggestion. Our assumption is based on the fact that deep neural networks trained with cross-entropy not only learn decision boundaries, but also implicitly estimate how likely an input belongs to the underlying data distribution (i.e., posterior confidence) (Zhang, 2004). As a result, a model tends to produce higher confidence for samples that align with its own training distribution, while assigning lower confidence for unseen or distribution-shifted samples such as adversarial inputs. This inherent behavior naturally results in a confidence separation between the two branches, eliminating the need for handcrafted detection components, and forms the theoretical foundation of our adversarial example detection that enables the method to detect adversarial examples without requiring any additional detection training.
>
> Furthermore, the empirical experiments illustrated in Figure 2 validate this claim. As shown in Figure 2(b), when evaluated on adversarial examples, the confidence of the BMP-branch is significantly lower than that of the MPAE-branch. This occurs because the MPAE-branch learns adversarial-specific data characteristics through adversarial training, while the BMP-branch does not.
>
> We have updated our manuscript to include the corresponding theoretical proof, and the complete proof has been provided in **Appendix C** as follows:
>
> #### Theoretical Analysis: Why BMP and MPAE Branches Exhibit Different Confidence
>
> The key rationale of ZeLDA is that neural networks naturally assign higher confidence to inputs that follow their training data distribution, and lower confidence to inputs that lie outside that distribution. This behavior is not manually designed, but emerges naturally from empirical risk minimization using cross-entropy. To support this statement, we present a theoretical proof as follows.
>
> #### Training Distributions of Two Branches
>
> Let $Q(x,y)$ denote the benign data distribution, where $x$ is the input image and $y \in$ {$1,\dots,K$} is the label. Then, $Q_\delta(x,y)$ represents the adversarial data distribution. Each sample has the form $x+\delta$ with $\|\delta\|_p \le \varepsilon$, where $\delta$ is the adversarial perturbation vector crafted by an attacker.
>
> Each branch is optimized on its own empirical training distribution, denoted as $P_b$ and $P_a$, respectively. In particular, the MPAE branch is trained on a mixture of benign and adversarial examples. Hence, $P_b$ and $P_a$ can be formulated as
>
> $$
> P_b(x,y) = Q(x,y),
> $$
>
> $$
> P_a(x,y) = (1-\alpha)\,Q(x,y) + \alpha\,Q_\delta(x,y), \quad \alpha \in (0,1),
> $$
>
> where $\alpha$ denotes the mixing coefficient that controls the proportion of adversarial examples used during training.
>
> Formally, $P_b$ serves as the empirical training distribution for the BMP branch, while $P_a$ represents the mixed training distribution for the MPAE branch, containing both benign and adversarial examples.
>
> #### Cross-Entropy Training and Posterior Approximation
>
> Consider a classifier $f(x)$ trained using the cross-entropy loss:
>
> $$
> \mathcal{L}(f) = \mathbb{E}_{(x,y)\sim P}[-\log f_y(x)],
> $$
>
> where $f_y(x)$ denotes the predicted probability for class $y$.
>
> Under sufficient model capacity and training convergence, the classifier trained on distribution $P(x,y)$ approximates the true posterior (Zhang, 2004), which can be formulated as
>
> $$
> f^\star(x) \approx P(y \mid x),
> $$
>
> where $f^\star$ refers to the theoretically optimal classifier. For our two branches, we have
>
> $$
> f_b(x) \approx P_b(y|x) \quad\text{and}\quad f_a(x) \approx P_a(y|x),
> $$
>
> where $f_b(x)$ and $f_a(x)$ denote the output probability vectors of the BMP branch and the MPAE branch, respectively.
>
> Then, we define the model confidence using the maximum softmax probability:
>
> $$
> m_b(x) = \max_k f_{b,k}(x) \approx \max_k P_b(y=k \mid x),
> $$
>
> $$
> m_a(x) = \max_k f_{a,k}(x) \approx \max_k P_a(y=k \mid x),
> $$
>
> where $f_{b,k}(x)$ (or $f_{a,k}(x)$) represents the predicted probability that input $x$ belongs to class $k$. $m_b$ and $m_a$ represent the confidence scores of the BMP and MPAE branches, respectively. Intuitively, $m_b(x)$ and $m_a(x)$ measure how well an input $x$ aligns with the data distributions $P_b$ and $P_a$, respectively.

---

> ### Author Response · Authors · 2025-11-23
> **Response to Reviewer 2cmk(2/4)**
>
> #### Model Sensitivity
>
> Then, for benign samples $x \sim Q(x,y)$, BMP's posterior is $P_b(y|x) = Q(y|x)$, while MPAE's posterior is diluted by the adversarial mixture $P_a(y|x) = (1-\alpha) Q(y|x) + \alpha Q_\delta(y|x)$. Obviously, for $x \sim Q(x,y)$, $Q(y|x)>Q_\delta(y|x)$. Hence, we have
>
> $$
> \begin{aligned}
> P_a(y|x)
> &= (1-\alpha)Q(y|x) + \alpha Q_\delta(y|x)< Q(y|x) = P_b(y|x).
> \end{aligned}
> $$
>
> As proved above, we conclude that for benign samples, the confidence of the BMP branch is strictly higher than that under the confidence of the MPAE branch. Conversely, for adversarial inputs, the MPAE branch yields a higher confidence than the BMP branch. This confirms a clear confidence separation between the two branches, demonstrating the inherent sensitivity of neural networks to shifts in data distributions. Specifically, a network tends to assign higher confidence to samples that align with its training distribution, while assigning lower confidence to unseen or out-of-distribution samples.
>
>
> ---
>
> > **W2:** The first and second claimed contributions can be merged for conciseness and clarity.
>
> **Response:** Thank you for your suggestion. We have merged them into one contribution in our revised manuscript.
>
> ---
>
> > **W3:**  The paper inconsistently uses adversarial example and adversarial example.
>
> **Response:** Thank you for your suggestion. We have unified the terminology and now consistently use "adversarial example" throughout the paper.
>
> ---
>
> > **W4:**  The Related Work section remains too high-level and lacks discussion of key concepts.
>
> **Response:** Thanks for your comment. Due to page limitations in the main text, we provided only a high-level discussion of related work to maintain focus and ensure the core method remains clear to readers. To address this concern, we have now included a substantially expanded and detailed review of the relevant concepts in our **Appendix A**, including pre-trained encoders, the pre-training paradigm, self-supervised learning approaches (e.g., SimCLR, MoCo), and DAE-based methods (e.g., PAP, AdvEncoder).
>
> ---
>
> > **W5:**  The Threat Model is essential for any security-related study and should appear in the main text rather than the appendix.
>
> **Response:** Thank you for your suggestion. In the revised manuscript, we have moved a concise version of the threat model and the key challenges to the main text to improve the logical flow and readability, while keeping the complete threat model in the appendix for reference.
>
> ---
>
> > **W6:**  The reported RA results (Tables 2 and 4) show large discrepancies compared to baseline values. The causes of these gaps should be clarified.
>
> **Response:** Thank you for your suggestions. The relatively large gaps in RA mainly occur because the baseline results correspond to the standard, undefended setting. Our results are consistent with observations reported in prior work on downstream-agnostic attacks, which have demonstrated strong attack transferability and thus typically result in low RA under such baseline conditions. This further reveals the vulnerability of existing models to downstream-agnostic adversarial perturbations and underscores the necessity of developing defenses against such attacks.
>
> As shown in Table 7 of our main text, other defense methods are able to significantly reduce the attack success rate compared to the undefended baseline, but often at the cost of noticeable degradation in benign accuracy. In contrast, our method not only substantially improves RA but also enhances performance on benign samples, indicating a better robustness–accuracy trade-off.
>
> To ensure reproducibility, we have provided detailed experimental settings in the main manuscript, and we will release all source code and configuration files upon paper acceptance to support further research in this area. We have added the related discussions in our appendix.

---

> ### Author Response · Authors · 2025-11-23
> **Response to Reviewer 2cmk(3/4)**
>
> > **W7:**   The proposed method demonstrates higher time and storage costs than the baselines (Table S3, L899–916).
>
> **Response:** Thank you for your comment. Our method increases the average inference time per image by only 0.00007 seconds. We admit that this represents a 30\% increase compared to the baselines, but the additional latency remains acceptable. Considering the substantial improvements in both BA and RA (over 30\% and 80\% respectively in some cases), we believe that this additional computational cost is justified and acceptable. In our future work, we will focus on reducing the computational overhead while further preserving both robustness and benign accuracy.
>
> While our method introduces multiple branches and therefore incurs a higher training cost per run, it only requires a single adversarial training process for different downstream tasks. In contrast, existing methods typically require separate adversarial training for each downstream dataset. Hence, in practical scenarios involving multiple downstream tasks, our approach can be more efficient overall. In particular, when the number of downstream tasks exceeds the number of branches, the total training cost of our method becomes lower than that of task-specific adversarial training.
>
> ---
>
> > **W8:**    The overall writing quality requires improvement, particularly in table captions, many of which contain grammatical errors. For instance, "Table 1: BA(\%) Baseline vs. ZeLAD in the semi-black-box scenario" is ungrammatical and should be revised for correctness and clarity.
>
> **Response:**  Thank you for the suggestion. We have carefully proofread the manuscript and revised typos in the manuscript.
>
>
> ---
>
> > **Q1:** Extenstion experiments to multimodal pre-trained models such as CLIP.
>
> **Response:** Thank you for your suggestion.We have added related experiments about the multimodal pre-trained models in **Appendix X**. As shown in the experiments, even with CLIP, a large-scale multimodal pre-trained encoder, our method maintains benign accuracy while achieving a substantial improvement in adversarial robustness. These findings demonstrate the strong generalizability of our approach to multimodal pre-trained encoders. The underlying reason is that the fundamental threat model of downstream-agnostic adversarial attacks remains consistent across both single-modality and multimodality encoders. Moreover, the inherent sensitivity leveraged by our defense strategy depends not on the specific pre-training modality, but on the degree of distribution alignment between the input samples and the encoder’s training data.
>
>
>
>
> **Table 1. BA (%) comparison of Baseline and ZePAD with CLIP pre-trained encoder.**
>
> | Dataset   | Method   | W-MSE | SimCLR | NNCLR | MoCo2+ | MoCo3 | AVG   |
> | --------- | -------- | ----- | ------ | ----- | ------ | ----- | ----- |
> | CIFAR10   | Baseline | 90.17 | 93.56  | 93.77 | 95.03  | 94.71 | 93.45 |
> |           | ZePAD    | 93.95 | 92.63  | 94.31 | 95.22  | 94.23 | 94.07 |
> | STL10     | Baseline | 78.49 | 81.94  | 83.09 | 84.46  | 84.00 | 82.40 |
> |           | ZePAD    | 95.07 | 96.64  | 97.33 | 97.36  | 97.68 | 96.82 |
> | ANIMALS10 | Baseline | 80.39 | 90.82  | 91.22 | 88.03  | 88.05 | 87.70 |
> |           | ZePAD    | 99.33 | 98.24  | 99.17 | 98.89  | 99.45 | 99.02 |

---

> ### Author Response · Authors · 2025-11-23
> **Response to Reviewer 2cmk(4/4)**
>
> **Table 2. RA (%) comparison of Baseline and ZePAD with CLIP pre-trained encoder.**
>
> | Dataset   | Method   | W-MSE | SimCLR | NNCLR | MoCo2+ | MoCo3 | AVG   |
> | --------- | -------- | ----- | ------ | ----- | ------ | ----- | ----- |
> | CIFAR10   | Baseline | 10.09 | 47.13  | 10.71 | 45.01  | 36.86 | 29.96 |
> |           | ZePAD    | 79.99 | 81.24  | 82.35 | 79.34  | 78.66 | 80.32 |
> | STL10     | Baseline | 41.16 | 67.52  | 44.05 | 65.55  | 64.28 | 56.51 |
> |           | ZePAD    | 92.65 | 91.73  | 92.78 | 91.04  | 92.33 | 92.11 |
> | ANIMALS10 | Baseline | 48.31 | 63.48  | 49.23 | 63.03  | 60.32 | 56.87 |
> |           | ZePAD    | 97.26 | 96.63  | 97.28 | 95.34  | 97.33 | 96.77 |
>
>
>
>
>
> ---
>
> > **Q2:** It is also unclear whether the method can generalize beyond image classification, for example to image retrieval, semantic segmentation, or object detection.
>
> **Response:** Thank you for your question. Our method is developed to defend downstream-agnostic encoder attacks, where the pre-trained encoder primarily supports classification-oriented tasks. While extending it to tasks such as detection or segmentation is theoretically possible, these tasks rely on fundamentally different architectural pipelines (e.g., U-Net variants) and make extensive use of skip connections or task-specific modules. Directly replacing their encoders with a standalone pre-trained encoder would likely lead to suboptimal performance, and thus, this direction is beyond the current scope of this paper. We appreciate this insightful suggestion and intend to explore extending our method to segmentation models, such as SAM, as part of future research.
>
> Nevertheless, to further validate the effectiveness of our framework, we extend our method to a retrieval task, where the encoder is directly used for feature extraction. As shown below, our method successfully defends against downstream-agnostic attacks in retrieval, demonstrating its effectiveness beyond classification tasks. These results further support the general applicability of our approach beyond classification-oriented tasks. The related discussions and results have been included in **Appendix J**.
>
>
>
> **Table 3. ZePAD's performance on the retrival task.**
>
> | Metric | method   | W-MSE | BYOL  | NNCLR | SimCLR | MoCo2+ | MoCo3 | ReSSL | DINO  | SwAV  | VibCreg | SupCon | AVG   |
> | ------ | -------- | ----- | ----- | ----- | ------ | ------ | ----- | ----- | ----- | ----- | ------- | ------ | ----- |
> | BA (%) | baseline | 62.25 | 62.19 | 58.85 | 60.10  | 69.05  | 70.22 | 62.89 | 56.32 | 69.73 | 75.47   | 59.74  | 64.26 |
> |        | zelad    | 90.49 | 86.74 | 84.53 | 88.79  | 89.00  | 90.20 | 78.97 | 80.37 | 81.00 | 95.26   | 88.85  | 86.75 |
> | RA (%) | baseline | 50.64 | 58.41 | 51.25 | 39.37  | 67.42  | 64.80 | 57.38 | 67.17 | 61.12 | 64.70   | 53.13  | 57.76 |
> |        | zelad    | 75.13 | 80.93 | 77.03 | 80.34  | 74.71  | 73.91 | 68.71 | 71.89 | 81.82 | 87.92   | 85.78  | 78.02 |
>
>
>
>
>  ## References
>
> Zhou, Z., Li, M., Liu, W., Hu, S., Zhang, Y., Wan, W., Xue, L., Zhang, L. Y., Yao, D., & Jin, H. (2024). Securely fine-tuning pre-trained encoders against adversarial examples. In *IEEE Symposium on Security and Privacy (SP)* (pp. 3015–3033). IEEE.
>
> Zhang, T. (2004). Statistical behavior and consistency of classification methods based on convex risk minimization. *The Annals of Statistics*, 32(1), 56–85.

---

> > ### Comment · Reviewer_2cmk · 2025-11-24
> > **Official Comment From Reviewer 2cmk**
> >
> > Thanks for the authors’ response. All of my concerns have been well addressed. This is a highly valuable piece of work, and I recommend acceptance. That said, there are still some writing issues, and I hope the authors will revise the paper according to the suggestions from all reviewers. I will raise my score to 8. Good luck !

---

> > > ### Author Response · Authors · 2025-11-24
> > > **Response to Reviewer 2cmk**
> > >
> > > Thank you very much for your positive and encouraging feedback. We are glad that our responses have addressed your concerns. We will carefully revise the paper and further polish the writing based on the suggestions from all reviewers. Thank you again for your time and support!

---

### Official Review · Reviewer_vmRw · 2025-10-31

**Soundness:** 3
**Presentation:** 3
**Contribution:** 3
**Rating:** 6
**Confidence:** 3

**Summary:**

This paper introduces **ZeLAD (Zero-Sacrifice Lifelong Adversarial Defense)**, a framework aimed at defending pre-trained self-supervised encoders against **Downstream-Agnostic Adversarial Examples (DAEs)**. Unlike prior task-specific fine-tuning approaches, ZeLAD claims to provide a single, lifelong adversarial defense applicable across diverse downstream tasks while maintaining or improving benign accuracy.

ZeLAD employs a dual-branch architecture:
- MPAE-Branch (Multi-Pattern Adversarial Enhancement Branch): Combines multiple adversarially fine-tuned pre-trained encoders trained with diverse SSL methods to enhance robustness through representational diversity.
- BMP-Branch (Benign Memory Preservation Branch): Trained solely on clean data to preserve benign performance.

During inference, a Robust Federal Decision Mechanism (RFDM) fuses branch outputs by comparing confidence scores. The model also demonstrates the ability to detect adversarial samples based purely on branch confidence disparities, without explicit adversarial detection training.

**Strengths:**

1. **Novelty and Conceptual Contribution:**
- The paper introduces the idea of “zero-sacrifice lifelong adversarial defense”, reframing adversarial robustness as a feature combination problem rather than a tradeoff problem.
- The dual-branch design (MPAE + BMP) and the federated confidence fusion mechanism are novel and well-motivated.
2. **Comprehensive Empirical Evaluation:**
- Extensive experiments across multiple SSL encoders (e.g., SimCLR, BYOL, MoCo, DINO) and datasets (CIFAR10, ImageNet, STL10, etc.) demonstrate the generality of the method.
- Results consistently show significant improvements over baselines and other defenses (e.g., TRADES, MART, Gen-AF).
3. **Strong Practical Relevance:**
- Addresses a genuine and underexplored problem: the vulnerability of pre-trained encoders to DAEs in a downstream-agnostic setting.
- The claim of “single tuning for all downstream tasks” has significant implications for scalable and resource-efficient deployment.
4. **Adversarial Detection Without Supervision:**
- The method’s ability to detect adversarial samples using confidence asymmetry without explicit training is an elegant byproduct.
5. **Clarity and Organization:**
- The paper is generally well-written and logically structured, with helpful figures (e.g., the overall ZeLAD architecture diagram and confidence distributions).

**Weaknesses:**

1. **Methodological Clarity and Rigor:**
- While conceptually interesting, some mathematical formulations (e.g., hybrid loss and cosine distance adjustment) lack detailed derivations and theoretical justification.
- The Robust Federal Decision Mechanism (RFDM) is empirically defined, but its weighting function (Eq. 8) seems heuristic and not theoretically grounded.
- There is no formal analysis of why confidence alignment is a robust signal or how it generalizes across tasks.

2. **Evaluation Limitations:**
- Most experiments focus on classification tasks; it’s unclear whether ZeLAD extends to non-classification downstream tasks (e.g., segmentation, detection).
- The adversarial attack diversity could be further improved—results rely heavily on AdvEncoder; other recent black-box attacks are not deeply explored.

3. **Scope of “Lifelong” Claim:**
- The “lifelong” aspect mainly refers to single fine-tuning across multiple tasks rather than continuous adaptation. There is no evidence of incremental task adaptation or continual learning capability, making the “lifelong” terminology somewhat overstated.

4. **Computational Cost and Practicality:**
- Maintaining multiple encoders (two adversarially fine-tuned and one benign) increases inference-time complexity and memory usage, which could hinder scalability.

**Questions:**

**1. Clarification on the “Lifelong” Claim**
You define ZeLAD as a *“lifelong adversarial defense”* requiring only a single tuning. However, lifelong learning typically implies *continuous adaptation to new tasks* without full retraining.
**Could you clarify how ZeLAD satisfies the lifelong learning property beyond single multi-task applicability?**
Have you tested whether ZeLAD can handle incremental task addition or domain shifts without catastrophic forgetting?

 **2. Justification for the “Zero-Sacrifice” Property**
The paper claims that ZeLAD achieves robustness improvements *without sacrificing benign accuracy*.
**What theoretical or empirical evidence supports this “zero-sacrifice” claim?**
Does this property hold under stronger perturbation budgets or adaptive attacks that target both branches simultaneously?

 **3. Robustness of the Confidence-Based Fusion (RFDM)**
The Robust Federal Decision Mechanism (RFDM) fuses branch outputs using confidence-based weighting.
**How reliable is this mechanism under confidence miscalibration, label noise, or adaptive attacks explicitly designed to manipulate confidence distributions?**
Have you compared this heuristic to alternative fusion schemes (e.g., entropy weighting, temperature scaling)?

 **4. Computational and Practical Efficiency**
ZeLAD employs multiple encoders (two adversarially fine-tuned and one benign), which could increase inference cost.
**What is the computational and memory overhead of ZeLAD compared to single-encoder fine-tuning methods?**
Can you comment on its scalability to larger SSL encoders (e.g., ViT-L/16, CLIP) in real-world applications?

~PS: Any chance you want to name the paper ZeLDA: Zero sacrifice Lifelong Defence against Adversaries?~

---

> ### Author Response · Authors · 2025-11-23
> **Response to Reviewer vmRW(1/5)**
>
> We sincerely appreciate the reviewer’s careful and constructive comments. Below, we provide detailed responses to each concern.
>
> > **W1.1:** While conceptually interesting, some mathematical formulations lack detailed derivations and theoretical justification.
>
> **Response:** Thank you for your comment. The hybrid loss and cosine distance adjustment are empirically motivated designs commonly used to balance the trade-off between robustness and benign accuracy. While they are not derived from strict theoretical optimality, they are widely adopted in adversarial training (Zhou et al., 2024) as effective mechanisms for stabilizing training and mitigating robustness–accuracy conflict.
>
> We provide a sensitivity analysis of the hyperparameter $\lambda$ in the appendix, and the corresponding results are shown below. From these results, we observe that when $\lambda$ is small, the RA remains low, indicating that the encoder is insufficiently adversarially tuned. As $\lambda$ increases, the model progressively balances BA and RA, achieving a better trade-off. We therefore adopt $\lambda = 20$ as the default setting, as it provides stable performance across different settings. The detailed discussions can be found in **Appendix G.5**.
>
>
>
> **Table 1. Ablation study on the effect of $\lambda$.**
>
> | $\lambda$ | W-MSE BA(%) | W-MSE RA(%) | SupCon BA(%) | SupCon RA(%) | SimCLR BA(%) | SimCLR RA(%) | NNCLR BA(%) | NNCLR RA(%) | BYOL BA(%) | BYOL RA(%) |
> | --------- | ----------- | ----------- | ------------ | ------------ | ------------ | ------------ | ----------- | ----------- | ---------- | ---------- |
> | 1         | 93.02       | 45.19       | 93.04        | 49.63        | 92.74        | 41.43        | 92.70       | 55.00       | 92.98      | 41.29      |
> | 10        | 93.37       | 87.15       | 92.46        | 76.84        | 93.70        | 74.30        | 93.53       | 79.53       | 93.18      | 81.70      |
> | 15        | 93.59       | 87.59       | 92.59        | 76.77        | 94.53        | 74.21        | 93.78       | 79.47       | 93.47      | 81.12      |
> | 20        | 93.42       | 86.84       | 92.39        | 77.99        | 95.05        | 74.70        | 93.96       | 79.84       | 93.49      | 80.18      |
> | 25        | 93.21       | 87.62       | 92.46        | 77.37        | 94.66        | 74.63        | 93.69       | 79.03       | 93.59      | 81.36      |
>
>
> ---
>
> > **W1.2 and Q3:** Theoretical analysis and related experiments about RFDM.
>
> **Response:** Thank you for your comment. Our approach is specifically designed to defend against downstream-agnostic attacks and has shown robust and transferable performance across various SSL encoders and datasets. The basic assumption underlying this work is the inherent sensitivity of neural networks, that is, models tend to assign higher confidence to samples that are consistent with their own training distribution.
>
> To effectively leverage this sensitivity, we design a confidence-based weighting mechanism in which each branch contributes to the final judgment in proportion to its confidence. Specifically, we adopt an exponential weighting function. The exponential form offers two key advantages:
>
> 1. It amplifies the difference between confidence scores, especially when one branch exhibits substantially higher certainty.
> 2. As the confidence score approaches 1, the corresponding weight increases sharply, allowing that branch to dominate the final decision. Such behavior is well aligned with our objective of allowing the branch that is most consistent with the sample’s underlying distribution to prevail.
>
> Considering that different attacks follow different threat models, directly applying our method to other types of attacks may not be straightforward or theoretically justified. However, we believe that extending this work to other types of attacks would represent a promising direction for future research.
>
> Furthermore, to empirically validate the effectiveness of the proposed RFDM, we conducted a series of experiments in the revised manuscript. Specifically, we compare RFDM with other ensemble strategies, including average- and entropy-ensemble methods, and the results are shown below. It can be noticed that RFDM achieves the highest RA in all settings, which demonstrates the effectiveness of our proposed method. Moreover, the performance of RFDM on BA is nearly identical to that of the average ensemble. This is because, as illustrated in Figure 2 of the main text, the confidence scores of the three branches for benign samples are nearly identical, in which case RFDM becomes approximately equivalent to the average-ensemble method. Detailed discussions can be found in **Appendix G.3**.

---

> ### Author Response · Authors · 2025-11-23
> **Response to Reviewer vmRW(2/5)**
>
> **Table 2. Performance under different ensemble methods.**
>
> | Ensemble Method | Metric | W-MSE | BYOL  | NNCLR | SimCLR | MoCo2+ | AVG   |
> | --------------- | ------ | ----- | ----- | ----- | ------ | ------ | ----- |
> | **Average**     | BA (%) | 93.43 | 93.61 | 93.91 | 93.95  | 93.57  | 93.69 |
> |                 | RA (%) | 85.46 | 79.85 | 77.58 | 71.93  | 64.38  | 75.84 |
> | **Entropy**     | BA (%) | 90.97 | 90.32 | 92.07 | 91.60  | 80.36  | 89.06 |
> |                 | RA (%) | 23.13 | 24.06 | 24.39 | 26.46  | 30.98  | 25.80 |
> | **RFDM**        | BA (%) | 93.37 | 93.49 | 93.69 | 93.70  | 93.57  | 93.56 |
> |                 | RA (%) | 87.14 | 81.70 | 79.66 | 74.30  | 65.91  | 77.74 |
>
> ---
>
> > **W1.3:** There is no formal analysis of why confidence alignment is a robust signal or how it generalizes across tasks.
>
> **Response:** Thank you for your suggestion. Our assumption is based on the fact that deep neural networks trained with cross-entropy not only learn decision boundaries, but also implicitly estimate how likely an input belongs to the underlying data distribution (i.e., posterior confidence) (Zhang, 2004). As a result, a model tends to produce higher confidence for samples that align with its own training distribution, while assigning lower confidence for unseen or distribution-shifted samples such as adversarial inputs. This inherent behavior naturally results in a confidence separation between the two branches, eliminating the need for handcrafted detection components, and forms the theoretical foundation of our adversarial example detection that enables the method to detect adversarial examples without requiring any additional detection training.
>
> Furthermore, the empirical experiments illustrated in Figure 2 validate this claim. As shown in Figure 2(b), when evaluated on adversarial examples, the confidence of the BMP-branch is significantly lower than that of the MPAE-branch. This occurs because the MPAE-branch learns adversarial-specific data characteristics through adversarial training, while the BMP-branch does not.
>
> We have updated our manuscript to include the corresponding theoretical proof, and the complete proof has been provided in **Appendix C** as follows:
>
> #### Theoretical Analysis: Why BMP and MPAE Branches Exhibit Different Confidence
>
> The key rationale of ZeLDA is that neural networks naturally assign higher confidence to inputs that follow their training data distribution, and lower confidence to inputs that lie outside that distribution. This behavior is not manually designed, but emerges naturally from empirical risk minimization using cross-entropy. To support this statement, we present a theoretical proof as follows.
>
> #### Training Distributions of Two Branches
>
> Let $Q(x,y)$ denote the benign data distribution, where $x$ is the input image and $y \in$ {$1,\dots,K$} is the label. Then, $Q_\delta(x,y)$ represents the adversarial data distribution. Each sample has the form $x+\delta$ with $\|\delta\|_p \le \varepsilon$, where $\delta$ is the adversarial perturbation vector crafted by an attacker.
>
> Each branch is optimized on its own empirical training distribution, denoted as $P_b$ and $P_a$, respectively. In particular, the MPAE branch is trained on a mixture of benign and adversarial examples. Hence, $P_b$ and $P_a$ can be formulated as
>
> $$
> P_b(x,y) = Q(x,y),
> $$
>
> $$
> P_a(x,y) = (1-\alpha)\,Q(x,y) + \alpha\,Q_\delta(x,y), \quad \alpha \in (0,1),
> $$
>
> where $\alpha$ denotes the mixing coefficient that controls the proportion of adversarial examples used during training.
>
> Formally, $P_b$ serves as the empirical training distribution for the BMP branch, while $P_a$ represents the mixed training distribution for the MPAE branch, containing both benign and adversarial examples.

---

> ### Author Response · Authors · 2025-11-23
> **Response to Reviewer vmRW(3/5)**
>
> #### Cross-Entropy Training and Posterior Approximation
>
> Consider a classifier $f(x)$ trained using the cross-entropy loss:
>
> $$
> \mathcal{L}(f) = \mathbb{E}_{(x,y)\sim P}[-\log f_y(x)],
> $$
>
> where $f_y(x)$ denotes the predicted probability for class $y$.
>
> Under sufficient model capacity and training convergence, the classifier trained on distribution $P(x,y)$ approximates the true posterior (Zhang, 2004), which can be formulated as
>
> $$
> f^\star(x) \approx P(y \mid x),
> $$
>
> where $f^\star$ refers to the theoretically optimal classifier. For our two branches, we have
>
> $$
> f_b(x) \approx P_b(y|x) \quad\text{and}\quad f_a(x) \approx P_a(y|x),
> $$
>
> where $f_b(x)$ and $f_a(x)$ denote the output probability vectors of the BMP branch and the MPAE branch, respectively.
>
> Then, we define the model confidence using the maximum softmax probability:
>
> $$
> m_b(x) = \max_k f_{b,k}(x) \approx \max_k P_b(y=k \mid x),
> $$
>
> $$
> m_a(x) = \max_k f_{a,k}(x) \approx \max_k P_a(y=k \mid x),
> $$
> where $f_{b,k}(x)$ (or $f_{a,k}(x)$) represents the predicted probability that input $x$ belongs to class $k$. $m_b$ and $m_a$ represent the confidence scores of the BMP and MPAE branches, respectively. Intuitively, $m_b(x)$ and $m_a(x)$ measure how well an input $x$ aligns with the data distributions $P_b$ and $P_a$, respectively.
>
> #### Model Sensitivity
>
> Then, for benign samples $x \sim Q(x,y)$, BMP's posterior is $P_b(y|x) = Q(y|x)$, while MPAE's posterior is diluted by the adversarial mixture $P_a(y|x) = (1-\alpha) Q(y|x) + \alpha Q_\delta(y|x)$. Obviously, for $x \sim Q(x,y)$, $Q(y|x)>Q_\delta(y|x)$. Hence, we have
>
> $$
> \begin{aligned}
> P_a(y|x)
> &= (1-\alpha)Q(y|x) + \alpha Q_\delta(y|x)< Q(y|x) = P_b(y|x).
> \end{aligned}
> $$
>
> As proved above, we conclude that for benign samples, the confidence of the BMP branch is strictly higher than that under the confidence of the MPAE branch. Conversely, for adversarial inputs, the MPAE branch yields a higher confidence than the BMP branch. This confirms a clear confidence separation between the two branches, demonstrating the inherent sensitivity of neural networks to shifts in data distributions. Specifically, a network tends to assign higher confidence to samples that align with its training distribution, while assigning lower confidence to unseen or out-of-distribution samples.
>
> ---
>
>
> > **W3 and Q1:** Concerns regarding the term "lifelong".
>
> **Response:**  Thank you for your comment. We understand that the term "lifelong" may lead to potential misunderstanding with the concept of continual learning. In our paper, "lifelong" does not refer to incremental task adaptation, but rather to the ability of a single adversarially trained model to operate across different downstream tasks without retraining or task-specific fine-tuning.
>
> In contrast to existing downstream-agnostic adversarial defense methods, which require separate adversarial training for each downstream task, our method performs adversarial training only once on a single dataset, after which it can be directly transferred to unseen downstream tasks while maintaining robustness. This "train-once, defend-forever" property is what we refer to as lifelong in our context.
>
> To avoid confusion, we have revised the term to "Persistent-Robustness" to avoid unnecessary misunderstandings.
>
> ---
>
> > **W2.1:** Most experiments focus on classification tasks; it's unclear whether ZeLAD extends to non-classification downstream tasks (e.g., segmentation, detection).
>
> **Response:** Thank you for your question.  Our method is developed to defend downstream-agnostic encoder attacks, where the pre-trained encoder primarily supports classification-oriented tasks. While extending it to tasks such as detection or segmentation is theoretically possible, these tasks rely on fundamentally different architectural pipelines (e.g., U-Net variants) and make extensive use of skip connections or task-specific modules. Directly replacing their encoders with a standalone pre-trained encoder would likely lead to suboptimal performance, and thus, this direction is beyond the current scope of this paper. We appreciate this insightful suggestion and intend to explore extending our method to segmentation models, such as SAM, as part of future research.
>
> Nevertheless, to further validate the effectiveness of our framework, we extend our method to a retrieval task, where the encoder is directly used for feature extraction. As shown below, our method successfully defends against downstream-agnostic attacks in retrieval, demonstrating its effectiveness beyond classification tasks. These results further support the general applicability of our approach beyond classification-oriented tasks. The related discussions and results have been included in **Appendix J**.

---

> ### Author Response · Authors · 2025-11-23
> **Response to Reviewer vmRW(4/5)**
>
> **Table 3. ZePAD's performance on the retrival task.**
>
> | Metric | method   | W-MSE | BYOL  | NNCLR | SimCLR | MoCo2+ | MoCo3 | ReSSL | DINO  | SwAV  | VibCreg | SupCon | AVG   |
> | ------ | -------- | ----- | ----- | ----- | ------ | ------ | ----- | ----- | ----- | ----- | ------- | ------ | ----- |
> | BA (%) | baseline | 62.25 | 62.19 | 58.85 | 60.10  | 69.05  | 70.22 | 62.89 | 56.32 | 69.73 | 75.47   | 59.74  | 64.26 |
> |        | ZeLAD    | 90.49 | 86.74 | 84.53 | 88.79  | 89.00  | 90.20 | 78.97 | 80.37 | 81.00 | 95.26   | 88.85  | 86.75 |
> | RA (%) | baseline | 50.64 | 58.41 | 51.25 | 39.37  | 67.42  | 64.80 | 57.38 | 67.17 | 61.12 | 64.70   | 53.13  | 57.76 |
> |        | ZeLAD    | 75.13 | 80.93 | 77.03 | 80.34  | 74.71  | 73.91 | 68.71 | 71.89 | 81.82 | 87.92   | 85.78  | 78.02 |
>
> ---
>
> > **W2.2:** The experiments rely heavily on AdvEncoder.
>
> **Response:** Thank you for your comment. We chose AdvEncoder because it represents the state-of-the-art adversarial attack method for pretrained encoders. In addition to AdvEncoder, we have also compared against other attack methods, including UAP, UAPGD, SSP, and PAP, as shown in Table 7 and Figure S1 in the Appendix. The results consistently demonstrate that our method maintains strong defensive performance under these additional attack types.
>
> Considering the page limitations and our extensive experimental settings involved, we were unable to include all black-box attack results under every scenario in the current submission. To ensure clarity and comparability, we reported the strongest attack method and provided representative results from other attacks. We sincerely appreciate this suggestion, and in the camera-ready version, we will strive to incorporate additional results to offer a more comprehensive analysis.
>
> ---
>
>
>  **Q2:** What theoretical or empirical evidence supports this ‘zero-sacrifice’ claim?
>
> **Response:** Thank you for your insightful question. Our method leverages an inherent property of neural networks: they tend to assign higher confidence to samples that align with their training data distribution (Zhang, 2004). Building on this, our proposed RFDM mechanism further amplifies the confidence discrepancy between the two branches, ensuring that for each input, the branch trained on the most relevant distribution naturally dominates the final prediction.
>
> This design yields two key benefits:
>
> 1. It guarantees a performance lower bound, as benign and adversarial examples are handled by the branch most sensitive to their corresponding distribution.
>
> 2. When both branches provide comparable predictions (e.g., under stronger or adaptive attacks), the fusion mechanism yields a more robust and balanced response rather than performance degradation.
>
>
> These benefits explain why our approach preserves benign accuracy while maintaining competitive robustness, a characteristic we refer to as the "zero-sacrifice" property in this paper.
>
> Additionally, we validate the effectiveness of our proposed method in a white-box attack scenario, and the results can be found in Tabs. S1 and S2 in our appendix. In this scenario, the attacker has full access to the model and can construct downstream-agnostic examples. Even in this challenging scenario, our method maintains strong and robust performance.
>
> Moreover, we further evaluate our method under stronger perturbation budgets, and the results are shown below.
> It can be noticed that, even when the attacker’s perturbation budget is increased by 60\%, the model still cannot be effectively compromised. The related experiments and discussions have been included in **Appendix G.4**.
>
>
> **Table 4. RA (\%) under different perturbation budgets.**
>
> | $\epsilon$ | BYOL  | ReSSL | SupCon | SimCLR | NNCLR |  AVG  |
> | :--------: | :---: | :---: | :----: | :----: | :---: | :---: |
> |   10/255   | 80.18 | 84.58 | 77.99  | 74.70  | 79.84 | 79.46 |
> |   12/255   | 74.29 | 72.41 | 71.83  | 61.80  | 70.72 | 70.21 |
> |   14/255   | 57.67 | 66.41 | 68.22  | 59.32  | 67.46 | 63.82 |
> |   16/255   | 53.49 | 66.53 | 61.17  | 54.45  | 53.11 | 57.75 |

---

> ### Author Response · Authors · 2025-11-23
> **Response to Reviewer vmRW(5/5)**
>
> > **Q4:** What is the computational and memory overhead of ZeLAD compared to single-encoder fine-tuning methods? Can you comment on its scalability to larger SSL encoders (e.g., ViT-L/16, CLIP) in real-world applications?
>
> **Response:**  Thank you for your comment.
> Our method increases the average inference time per image by only 0.00007 seconds. We admit that this represents a 30\% increase compared to the baselines, but the additional latency remains acceptable. Considering the substantial improvements in both BA and RA (over 30\% and 80\% respectively in some cases), we believe that this additional computational cost is justified and acceptable. In our future work, we will focus on reducing the computational overhead while further preserving both robustness and benign accuracy.
>
> While our method introduces multiple branches and therefore incurs a higher training cost per run, it only requires a single adversarial training process for different downstream tasks. In contrast, existing methods typically require separate adversarial training for each downstream dataset. Hence, in practical scenarios involving multiple downstream tasks, our approach can be more efficient overall. In particular, when the number of downstream tasks exceeds the number of branches, the total training cost of our method becomes lower than that of task-specific adversarial training.
>
> In addition, we add related experiments with multimodal pre-trained encoders in **Appendix X**. As shown in the experiments, even with CLIP, a large-scale multimodal pre-trained encoder, our method maintains benign accuracy while achieving a substantial improvement in adversarial robustness. These findings demonstrate the strong generalizability of our approach to multimodal pre-trained encoders. The underlying reason is that the fundamental threat model of downstream-agnostic adversarial attacks remains consistent across both single-modality and multimodality encoders. Moreover, the inherent sensitivity leveraged by our defense strategy depends not on the specific pre-training modality, but on the degree of distribution alignment between the input samples and the encoder’s training data.
>
>
>
> <div style="overflow-x: auto;">
>
>
> **Table 5. BA (%) comparison of Baseline and ZePAD with CLIP pre-trained encoder.**
>
> | Dataset   | Method   | W-MSE | SimCLR | NNCLR | MoCo2+ | MoCo3 | AVG   |
> | --------- | -------- | ----- | ------ | ----- | ------ | ----- | ----- |
> | CIFAR10   | Baseline | 90.17 | 93.56  | 93.77 | 95.03  | 94.71 | 93.45 |
> |           | ZePAD    | 93.95 | 92.63  | 94.31 | 95.22  | 94.23 | 94.07 |
> | STL10     | Baseline | 78.49 | 81.94  | 83.09 | 84.46  | 84.00 | 82.40 |
> |           | ZePAD    | 95.07 | 96.64  | 97.33 | 97.36  | 97.68 | 96.82 |
> | ANIMALS10 | Baseline | 80.39 | 90.82  | 91.22 | 88.03  | 88.05 | 87.70 |
> |           | ZePAD    | 99.33 | 98.24  | 99.17 | 98.89  | 99.45 | 99.02 |
>
> <div style="overflow-x: auto;">
>
>
> **Table 6. RA (%) comparison of Baseline and ZePAD with CLIP pre-trained encoder.**
>
> | Dataset   | Method   | W-MSE | SimCLR | NNCLR | MoCo2+ | MoCo3 | AVG   |
> | --------- | -------- | ----- | ------ | ----- | ------ | ----- | ----- |
> | CIFAR10   | Baseline | 10.09 | 47.13  | 10.71 | 45.01  | 36.86 | 29.96 |
> |           | ZePAD    | 79.99 | 81.24  | 82.35 | 79.34  | 78.66 | 80.32 |
> | STL10     | Baseline | 41.16 | 67.52  | 44.05 | 65.55  | 64.28 | 56.51 |
> |           | ZePAD    | 92.65 | 91.73  | 92.78 | 91.04  | 92.33 | 92.11 |
> | ANIMALS10 | Baseline | 48.31 | 63.48  | 49.23 | 63.03  | 60.32 | 56.87 |
> |           | ZePAD    | 97.26 | 96.63  | 97.28 | 95.34  | 97.33 | 96.77 |
>
>
>
>
> ---
>
> > **PS:** Any chance you want to name the paper ZeLDA: Zero sacrifice Lifelong Defence against Adversaries?
>
> **Response:** Thank you for the creative suggestion! We are all actually big fans of the Zelda series. Once we noticed the abbreviation could resemble "ZeLDA," we immediately loved it. However, since the term "lifelong" may be misleading and could be interpreted as referring to continual learning, we have revised the title to ensure clarity and precision.
>
>
>
>  ## References
>
> Zhou, Z., Li, M., Liu, W., Hu, S., Zhang, Y., Wan, W., Xue, L., Zhang, L. Y., Yao, D., & Jin, H. (2024). Securely fine-tuning pre-trained encoders against adversarial examples. In *IEEE Symposium on Security and Privacy (SP)* (pp. 3015–3033). IEEE.
>
> Zhang, T. (2004). Statistical behavior and consistency of classification methods based on convex risk minimization. *The Annals of Statistics*, 32(1), 56–85.

---

> > ### Comment · Reviewer_vmRw · 2025-11-24
> >
> > Thank you for addressing all the questions and weaknesses I had brought up. I am very happy with the response received and I have no further questions or doubts. With that said, I am increasing the score to an 8.

---

> > > ### Author Response · Authors · 2025-11-24
> > > **Response to Reviewer vmRw**
> > >
> > > Thank you for the positive and encouraging feedback. We are delighted that our responses have adequately addressed your concerns.

---

### Author Response · Authors · 2025-11-23
**General Response to PCs, SACs, ACs, and Reviewers**

Dear PCs, SACs, ACs, and Reviewers,

We sincerely appreciate your diligent efforts and insightful feedback. Your comments have greatly helped us improve the quality and clarity of our manuscript. To assist the newly assigned AC and help reduce the workload, we provide below a summary of the key points from the reviews and the reviewer-author discussions.

We are pleased to note that the reviewers recognized several strengths of our work, including:

- The novelty of our method and the substantial advancement of the concept (Reviewers `vmRw`, `gcvV`).

- The clarity and readability of the paper (Reviewer `2cmk`) and its logical organization (Reviewer `pqbZ`).

- The comprehensiveness of our experimental evaluations (Reviewers  `vmRw`, `gcvV`, `2cmk`, `pqbZ`).

- The strong practical relevance of our method (Reviewers `gcvV`, `vmRw`), along with the elegant emergence of adversarial exsample detection (Reviewer `vmRw`).

Following the valuable suggestions from the reviewers, we have provided detailed responses to all comments in the rebuttal PDF and made the following major improvements:

- Theoretical justifications and formal analysis were provided to support the proposed method.

- We conducted extensive experiments to analyze the impact of key hyperparameters and provided detailed descriptions of the experimental settings.

- Ablation studies are included to demonstrate the effectiveness of the proposed RFDM.

- We clarified the use of the term "lifelong" and revised it to avoid potential misunderstandings.

- We extended our defense framework to additional tasks beyond classification.

- We validated the generalizability of our approach using the multimodal pre-trained encoder CLIP.

- We expanded the discussion of related work in the Appendix.

- We carefully proofread the manuscript and corrected all typographical and mathematical writing issues.

---

**Recognition of our revision from reviewers.** All four reviewers explicitly acknowledged during the discussion that their concerns had been satisfactorily addressed, and three of them raised their scores accordingly. **Reviewer vmRW** increased the score from **6→8**, **Reviewer 2cmk** increased the score from **4→8**, and **Reviewer pdbZ** raised the score from **4→6**. **Reviewer gcvV** maintained the score of 6, confirming that the key concerns had been resolved.

As a result, the final score distribution before the rollback was: **8, 8, 6, and 6**, and these acknowledgments were explicitly documented in the discussion.

---

Above, we have faithfully summarized all reviewer comments and our corresponding responses, hoping that this will assist the newly assigned AC in evaluating our work.

We are deeply grateful to the reviewers, ACs, SACs and PCs for their time, insightful feedback, and dedicated efforts throughout this process. We believe that this review round has greatly improved the quality of our manuscript.

Best regards,

The Authors

---

### Meta-Review · Area_Chair_qdgC · 2026-01-07

**Summary:**

All four reviewers acknowledged that their concerns had been addressed, and three of them raised their scores accordingly. As clearly indicated by the reviewers’ comments, the final score distribution before the rollback was: 8, 8, 6, and 6.

**Reviewer Concerns:**

I believe the major concerns have been solved by the excellent rebuttal.

**Reviewer Scores:**

As clearly indicated by the reviewers’ comments, the final score distribution before the rollback was: 8, 8, 6, and 6.

---

### Decision · Program_Chairs · 2026-01-26

Accept (Poster)